# Learning to Cooperate with Humans through Theory-Informed Trust Beliefs

## Abstract

Real-world human–AI cooperation is challenging due to the wide range of interests and capabilities that each party brings. To maximize joint performance, cooperative AI must adapt its policies to the competence and incentives of its specific human partner. Prevailing approaches address this challenge by training on human data or simulated partners. In this paper, we pursue an orthogonal approach: grounded on theory from social science, we hypothesize that equipping agents with human-like trust beliefs enables them to adapt to human partners more effectively. We formulate the cooperative agent's problem as TRUSTPOMDP, a variant of POMDPs, and develop a trust model that captures three key factors known to shape human trust beliefs: ability, benevolence, and integrity (ABI). A key advantage of the approach is that it only requires minimal modifications to a POMDP agent. TRUSTPOMDPs can be trained with real or simulated partners, provided sufficient diversity in the three dimensions. Results from both simulated and human-subject experiments (N=106) show that TRUSTPOMDP-based agents adapt more rapidly and effectively to various partners, while baselines methods tend to over- or undertrust, reducing team performance. These findings highlight the promise of incorporating social science-informed trust models into RL agents to advance collaboration with humans.

## 1 Introduction

Cooperating with humans in real-world environments requires accounting for their diverse capabilities, motivations, and behaviors (Wang et al., 2024; Hong et al., 2023). Some human partners may have limited competence, others may prioritize personal credit over team success, and still others may be willing to violate social norms (Summerfield & Tsetsos, 2015; Cacioppe, 1999; Haselton et al., 2015). As illustrated in Figure 1, such factors should be accounted for, or the agent may risk waiting in vain for help from a selfish teammate, delegate critical tasks to an incompetent one, or rely on someone who disregards norms.

How to learn cooperative policies that adapt to such characteristics of human partners is an open problem. Prevailing approaches tackle this challenge by training on human data (Carroll et al., 2019) or on simulated partners (Carroll et al., 2019; Papoudakis et al., 2021; Liang et al., 2024; Hong et al., 2023; Strouse et al., 2021). Recent work on *zero-shot coordination* (ZSC) emphasizes generalization by exposing agents to diverse partners (Carroll et al., 2019; Papoudakis et al., 2021; Liang et al., 2024; Hong et al., 2023; Strouse et al., 2021), typically through constructing simulated partner populations with diversity (Papoudakis et al., 2021; Liang et al., 2024; Strouse et al., 2021).

In this paper, we take an orthogonal approach. We build on theories of trust from social and behavioral sciences. Correctly calibrated trust is a requirement for effective human collaboration (Mayer et al., 1995; Lewicki et al., 2006; McAllister, 1995; Cook et al., 2005). In our work, we want to exploit a key insight from this literature, which is that humans form and update *trust beliefs* about their partners, which in turn guide reliance, allocation of tasks, and strategies of cooperation, with positive effects on team performance (Dirks, 1999; De Jong et al., 2016). Informed by these findings, we hypothesized that *equipping agents with human-like trust beliefs will enable them to effectively adapt to diverse and previously unseen human partners*.

Our technical contribution is the definition and study of a novel variant of the Partially Observable Markov Decision Process (POMDP) that incorporates a belief model designed to capture three key

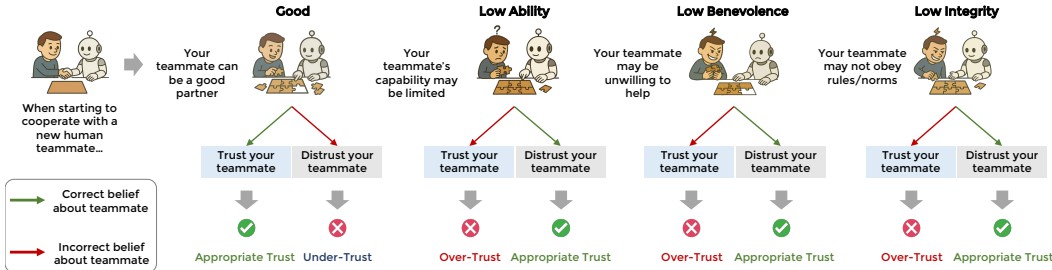

Figure 1: When cooperating with a human, optimal policy depends on how competent, benevolent, and norm-obeying the partner is. Learning an accurate representations about these factors enable an agent to adapt its policy better, whereas incorrect beliefs can lead to miscalibrated trust—either over-trusting (e.g., relying on an incapable or uncooperative partner) or under-trusting (e.g., failing to rely on a competent and well-intentioned partner).

traits that humans naturally consider in interpersonal collaboration (Mayer et al., 1995). *Ability* denotes the belief that the trustee has the competence to be effective, *Benevolence* the belief that the trustee intends to act in the trustor's interest beyond self-gain, and *Integrity* the belief that the trustee upholds principles and norms acceptable to the trustor (Mayer et al., 1995). We formalize TRUST-POMDP, in which a human partner's ABI traits are unobservable to the AI. The agent has a belief model that allows it to infer these traits probabilistically and to condition its policy accordingly. A notable advantage of this formulation is its representational efficiency: in the minimal setup examined in this paper, only two additional observation variables (the mean and uncertainty) per ABI dimension are added to a standard POMDP agent. We further prove when the human partner behaves as social science suggests (i.e., is ABI-like), the approach improves cooperative policies.

We propose *Trust Co-play*, an approach to training TRUSTPOMDPs inspired by work on ZSC. In principle, TRUSTPOMDPs can be trained with either real or simulated partners, provided there is sufficient diversity across the three dimensions. In our approach, we construct a trustee agent population by varying the ABI traits. We vary the levels of ability through Boltzmann rationality, while benevolence and integrity are controlled via reward design. This yields a controllable distribution of partner behaviors, ensuring that the agent learns to deal with extreme behaviors that may be more rare in human behavior but that require adapting one's policy (e.g., norm-abusing partners). Further, Trust Co-play allows training a probabilistic ABI inference model, which in turn allows the agent to better handle uncertainty and scarce observations.

We systematically evaluate the approach with synthetic and real humans in *Overcooked*, a widely used and complex multi-agent environment (Hong et al., 2023; Wang et al., 2024; Strouse et al., 2021; Zhao et al., 2023). First, in the simulation study, we compared TRUSTPOMDP with established ZSC methods—FCP (Strouse et al., 2021) and MEP (Zhao et al., 2023)—as well as an ablation baseline: a POMDP agent also trained on the trustee population but without the ABI model. TRUSTPOMDPs achieved on average higher team rewards. Second, we conducted a human-subject experiment ($N = 106$) in which participants were free to interact with the AI agents in any way they chose. TRUSTPOMDP again achieved the highest team rewards, adapting more effectively to diverse human partners and yielding a better cooperative experience. In contrast, in both studies, the baselines often exhibited miscalibrated trust—either over-trusting or under-trusting. Our findings highlight the promise of drawing from social sciences to build human-like inferential capabilities into cooperative agents that work with humans.

## 2 RELATED WORK

**Trust in Human-Human Collaboration**. Trust—defined as the willingness to be vulnerable based on positive expectations of another's behavior (Mayer et al., 1995)—is fundamental to human collaboration. It influences behavior in information sharing, joint problem solving, and tolerance for mistakes (McAllister, 1995; Lewicki et al., 2006), and plays a critical role in coordination, conflict resolution, and the pursuit of shared goals (Olson et al., 2006; Williams, 2001). Appropriately calibrated trust is essential for effective teamwork, whereas over-trust or under-trust can lead to

suboptimal or failed collaborative outcomes (Lee & Moray, 1994). Among existing trust theories, the ability–benevolence–integrity (ABI) model offers a compact account of interpersonal trust and explains diverse cooperative behaviors (Mayer et al., 1995) and has been extended and verified by many researchers (Yan & Holtmanns, 2008; Cho et al., 2015). Building on this theory, we extend trust modeling from human–human to human–AI collaboration, enabling AI agents to iteratively evaluate their partners' reliability and adapt their behaviors accordingly.

**Trust in Human-AI Cooperation**. In human–AI collaboration, human trust is influenced by factors such as AI capability (Yin et al., 2019; Rechkemmer & Yin, 2022), transparency (Zhang et al., 2020), explainability (Wang & Yin, 2021), and uncertainty communication (Schemmer et al., 2023; Ma et al., 2023; Bansal et al., 2021; Rastogi et al., 2022). Human trust in AI and automation has been studied for decades. For example, Chen et al. (2020) inferred human trust in a robot and adjusted the robot's policy to improve team performance, while Siu et al. (Siu et al., 2021) examined trust dynamics in human–AI teams in Hanabi. Related concepts such as *legibility* and *predictability* have also been shown to shape trust by improving the interpretability of agent behavior (Dragan et al., 2013). Lee's work on *trust in automation* further highlights the importance of designing systems that support appropriate reliance and calibrated trust (Lee & See, 2004). However, most prior work generally assumes a unidirectional form of trust in which humans are treated as trustworthy, positioning the human as the trustor and the AI as the trustee. In real-world cooperation, humans also vary in trustworthiness. Effective collaboration therefore requires *bidirectional trust*, where AI agents can evaluate the reliability of their human partners and learn when and how to trust them. This paper advances this underexplored perspective.

**Zero-shot Coordination (ZSC)**. A central goal of ZSC is to learn to coordinate effectively with previously unseen partners, whether other AI agents or humans (Wang et al., 2024; Carroll et al., 2019). Existing approaches can be grouped into three categories. (1) *Training with human data*. Some methods leverage datasets of human cooperation (Carroll et al., 2019), but these are limited in scale, subject to highly diverse human behaviors, and struggle to capture latent preferences, often resulting in brittle coordination policies (Hong et al., 2023). (2) *Inferring partner types*. Some papers adopt Theory of mind (Premack & Woodruff, 1978) approachs to infer latent partner traits using Bayesian models (Wu et al., 2021; Shum et al., 2019) or learned embeddings (Grover et al., 2018; Papoudakis et al., 2021), enabling adaptation to different types of partners. However, the inferred latent variables often lack interpretability. (3) *Zero-shot coordination via simulated populations*. Agents are trained with diverse simulated partners to improve generalization, using techniques such as FCP (Strouse et al., 2021), MEP (Zhao et al., 2023), LIPO (Charakorn et al., 2023), HSP (Yu et al., 2023b), TrajeDi (Lupu et al., 2021), and CoMeDi (Sarkar et al., 2023). These methods introduce variation in partners' abilities or preferences. Yet they overlook human *trustworthiness*—even though it plays a central role in collaboration. In contrast, we take an orthogonal approach: explicitly modeling a trust belief about human partners, grounded in established social science theory.

## 3 PRELIMINARY

**Partially Observable Markov Decision Process (POMDP).** A POMDP (Kaelbling et al., 1998) is defined as $\mathcal{M} = \langle \mathcal{S}, \mathcal{A}, \mathcal{O}, \mathcal{T}, \mathcal{R}, \gamma \rangle$, where the agent receives partial observations and acts to maximize expected discounted return.

**Human–AI Cooperative Game.** Human–AI cooperation is often modeled as a two-player POMDP with a shared team reward (Carroll et al., 2019; Strouse et al., 2021):

$$\mathcal{M} = \langle \mathcal{S}, \ \mathcal{A}_H, \ \mathcal{A}_A, \ \mathcal{O}_H, \ \mathcal{O}_A, \ \mathcal{T}, \ \mathcal{R}, \ \gamma \rangle,$$

where $\mathcal{S}$ is the state space; $\mathcal{O}_H, \mathcal{O}_A$ are the human and AI observation spaces; $\mathcal{A}_H, \mathcal{A}_A$ their action spaces; $\mathcal{T}$ the transition dynamics; $\mathcal{R}$ the team reward; and $\gamma$ the discount factor. The objective of *cooperative AI* is to learn a policy that maximizes expected return against diverse human partners:

$$\max_{\pi_A} \ \mathbb{E}_{\pi_H \sim P_H}[J(\pi_A, \pi_H)], \quad J(\pi_A, \pi_H) = \mathbb{E}\left[\sum_t \gamma^t \mathcal{R}(s_t, a_t^A, a_t^H)\right].$$

**Interactive POMDPs.** However, real-world cooperation often involves partially aligned rewards (Gallo Jr & McClintock, 1965). To deal with multi-agent settings with different (and possibly conflicting) objectives, researchers have proposed a general extension of POMDPs, known as Interactive

POMDPs (I-POMDPs) (Gmytrasiewicz & Doshi, 2004). In a two-agent setting (agent $i$ and agent $j$), an I-POMDP of agent $i$ is:

$$\text{I-POMDP}_i = \langle IS_i, A, T_i, \Omega_i, O_i, R_i \rangle$$

where $IS_i$ is a set of **interactive states** defined as $IS_i = S \times \Theta_j$, where $S$ is the set of states of the physical environment, and $\Theta_j$ is the set of possible intentional models of agent $j$. $A = A_i \times A_j$ is the finite set of joint actions. $T_i$ is a transition function, $T_i : IS_i \times A \times IS_i \to [0, 1]$, which describes the results of agents' actions. $\Omega_i$ is the set of agent $i$'s observations. $O_i : IS_i \times A \times \Omega_i \to [0, 1]$ is an observation function. Agent $i$'s reward $R_i$ is defined as, $R_i : IS_i \times A \to \mathbb{R}$.

The core idea of I-POMDPs is that an agent's belief is defined as a probability distribution over both the environmental states and the models of other agents. In this paper, our approach can be viewed as a simplified instantiation of the I-POMDP framework, in which the AI agent maintains beliefs over the state of the environment and the model of its human partner, represented by ABI (Ability, Benevolence, and Integrity), and makes decisions based on these beliefs.

Moreover, I-POMDPs offer a flexible framework for recursive belief modeling: not only can agent $i$ update its belief about agent $j$, but agent $j$ can, in principle, also update its belief about agent $i$. In such fully recursive settings, agent $i$ would need to anticipate how agent $j$ updates its beliefs in response to observed behaviors. In this work, however, we focus on a single-sided belief formulation, where the AI agent models the human partner but does not explicitly model the human's belief about the AI. This design choice is driven by our goal of creating a human-belief-agnostic collaborative agent, one that can robustly adapt to diverse human partners without relying on assumptions about their internal beliefs regarding the AI. Such a formulation better reflects real-world settings, where human beliefs are highly heterogeneous and often unobservable. We consider bidirectional belief modeling an important direction for future work.

## 4 METHOD

### 4.1 PROBLEM FORMULATION: TRUSTPOMDP

We model cooperation with a human partner who may vary in capability and pursue incentives only partially aligned with the AI's as a TRUSTPOMDP from the AI's perspective. The partner is characterized by a latent trustworthiness type (ABI) $z \in \mathcal{Z}$, which is unobservable to the AI and must be inferred through ongoing interaction (Figure 2b). Formally,

$$\mathcal{M}_{\text{TrustPOMDP}} = \langle \mathcal{S}, \mathcal{O}, \mathcal{Z}, \mathcal{A}, \mathcal{T}, \mathcal{U}, \mathcal{R}, \hat{\mathcal{Z}} \rangle,$$

where $\mathcal{S}$ is the environment state space; $\mathcal{O}$ the AI agent's observation space; $\mathcal{Z}$ the human partner's trustworthiness (ABI) space; $\mathcal{A}$ the AI agent's action space; $\mathcal{T}$ the transition dynamics under joint actions; $\mathcal{U}$ the inference function updating the belief $\hat{z}_t$ from interaction history; $\mathcal{R}$ the AI's reward function; and $\hat{\mathcal{Z}}$ the AI's ABI belief space ($\hat{z}_t \in \hat{\mathcal{Z}}$). The AI agent follows a

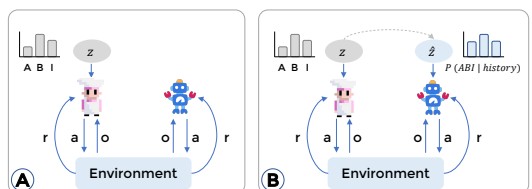

Figure 2: When collaborating with humans, AI agents encounter partners with varying ABI (Ability, Benevolence, Integrity) traits. (a) In a standard POMDP, the agent has no explicit representation of the human partner's latent ABI. (b) In TRUST-POMDP, the agent uses observations to form a belief over the human's latent ABI and incorporates it into its observations, enabling policies that better adapt to the human partner's traits.

trust-aware policy that conditions not only on its observation $o_t \in \mathcal{O}$ but also on its current belief $\hat{z}_t$ of the human partner's latent ABI state: $\pi^{\text{AI}}(a_t^{\text{AI}} \mid o_t, \hat{z}_t)$. We further show that TRUSTPOMDP preserves the Markov property of standard POMDPs in Appendix A.1.

### 4.2 MODELING ABI

With TRUSTPOMDP, we aim to equip the AI agent with the ability to infer a human partner's trustworthiness (Mayer et al., 1995). To this end, we construct a synthetic population of agents with diverse ABI profiles, grounded in trust theory—referred to as the *trustee agent population*. The modeling of each ABI dimension is detailed below.

**Ability: Rationality-Modulated Policy via Boltzmann Distribution**    Instead of encoding ability directly as an estimate of achievable reward, we model it by modulating policy stochasticity through *Boltzmann rationality* (Baker et al., 2007; Bobu et al., 2020) applied post-training. The policy of agent $i$ is defined as

$$\pi_i(a \mid s) = \frac{\exp(\beta_i Q_i(s, a))}{\sum_{a' \in \mathcal{A}} \exp(\beta_i Q_i(s, a'))}, \tag{1}$$

where $Q_i(s, a)$ is the action-value function, $\mathcal{A}$ is the action space, and $\beta_i \in [0, +\infty)$ is the rationality coefficient. Larger $\beta_i$ produces more rational, less stochastic behavior, reflecting higher ability. This formulation is agnostic to the agent's original reward, policy, or task, and enables systematic variation of ability.

In addition, there are alternative ways to model ability, such as manipulating an agent's observation range (Lieder & Griffiths, 2020), for instance through mechanisms like humans' size-limited foveal vision (Duchowski, 2018), introducing perceptual noise in observations (Sun et al., 2023), or constraining memory capacity (Mullainathan, 2002). However, the relationship between these factors and the resulting ability is often non-linear and lacks precise controllability. Ability can also be operationalized by selecting policies from different stages of RL training. For example, Fictitious Co-Play (FCP) (Strouse et al., 2021) uses early-stage checkpoints as low-ability agents. However, such approaches may inadvertently capture transient or inconsistent suboptimal strategies that do not reliably represent meaningful low-ability behavior. Therefore, we adopt the commonly used Boltzmann rationality framework to model ability (Laidlaw & Dragan, 2022).

**Benevolence: Partner-Oriented Reward via Event-Based Credit Assignment**    To model benevolence, drawing on social MDPs (Leibo et al., 2017) and credit-assignment methods in multi-agent RL (Zhou et al., 2020), we adjust how agents weight their own reward versus their partner's or shared reward. The benevolence-weighted reward of agent $i$ is defined as:

$$R_i^{(B)} = \alpha \cdot r_{\text{self}} + \beta \cdot r_{\text{other}} + (1 - \alpha - \beta) \cdot r_{\text{shared}}, \tag{2}$$

where $r_{\text{self}}$ denotes the reward obtained exclusively by agent itself, $r_{\text{other}}$ denotes the reward obtained exclusively by the partner agent, and $r_{\text{shared}}$ denotes the reward jointly shared by both agents. The parameters $\alpha$ and $\beta$ lie in $[0, 1]$ and satisfy $\alpha + \beta \le 1$.

In this work, we treat both increasing the partner's reward and increasing the shared reward as manifestations of benevolence. Therefore, benevolence can be effectively controlled using a single parameter $\alpha$. A larger $\alpha$ indicates a more self-oriented, low-benevolence agent, whereas a smaller $\alpha$ implies that the agent places greater emphasis on the partner's or shared reward, corresponding to a high-benevolence agent.

**Integrity: Norm Adherence via Reward Design**    Integrity is closely tied to adherence to social and ethical norms  (Mayer et al., 1995; Huberts, 2018). We model integrity by penalizing norm-violating actions. Formally, let $\mathcal{V}$ denote the set of norm-violating actions, which may be defined by the scenario through explicit task rules, social conventions, or imposed constraints. Agent $i$ then receives an integrity-related penalty:

$$R_i^{(I)} = \begin{cases} \delta, & \text{if } a_i \in \mathcal{V}, \\ 0, & \text{otherwise}, \end{cases} \tag{3}$$

where $\delta$ denotes the magnitude of the norm-violating incentive. A positive $\delta$ encourages unethical or deceptive behavior, reducing integrity, whereas a negative $\delta$ discourages norm-violating actions, fostering higher integrity.

## 4.3    Inferring ABI

While ABI dimensions can in principle vary continuously, without loss of generality, we simplify by discretizing each into binary values (0 for low, 1 for high). This still yields diverse policies through their interplay, though extending to finer-grained, continuous forms remains for future work.

To enable the trustor agent to infer its partner's ABI, we design an inference model that represents each dimension with a Beta distribution rather than a single scalar. The Beta distribution is well-suited for variables bounded in $[0, 1]$ and naturally models evidence accumulation (e.g., successes vs.

failures) (Nielsen et al., 2007), which aligns with the process of incremental trust updating during interaction. Moreover, the Beta distribution has been widely adopted in prior work for modeling human trust updating (Guo et al., 2021; Bhat et al., 2022; Chen et al., 2018; Guo & Yang, 2021; Dagdanov et al., 2025).

$$q_\phi(\hat{z}_d \mid x_{1:T}) = \text{Beta}(\alpha_d(x_{1:T}; \phi), \ \beta_d(x_{1:T}; \phi)), \quad d \in \{A, B, I\}, \tag{4}$$

where $x_{1:T}$ is the observed interaction history, $\hat{z}_d$ the inferred latent trust variable for dimension $d$, and $\phi$ the network parameters. The predictive mean and concentration are $p_d = \frac{\alpha_d}{\alpha_d + \beta_d}$, $S_d = \alpha_d + \beta_d$, with $p_d$ estimating ABI level and $S_d$ quantifying confidence. We prove the benefits of maintaining a trust belief when the human partner's ABI is uncertain in Appendix A.2.

### 4.4 TRAINING AND DEPLOYMENT OF THE BELIEF MODEL.

Each trustee agent in the population is annotated with a ground-truth ABI trait. We adopt a supervised approach. Given ground-truth ABI labels $y_d \in 0, 1$ for each dimension $d$, the model outputs Beta parameters $(\alpha_d, \beta_d)$ and we use the Beta mean $p_d = \frac{\alpha_d}{\alpha_d + \beta_d}$ as the predicted probability. The per-dimension loss combines a Bernoulli cross-entropy (BCE) term with an evidential regularizer that penalizes overconfident Beta shapes via a KL divergence to a uniform prior $\text{Beta}(1, 1)$:

$$\mathcal{L}_d = \underbrace{\text{BCE}(p_d, y_d)}_{\text{data fit}} + \lambda \cdot \underbrace{\text{KL}(\text{Beta}(\alpha_d, \beta_d) \,\|\, \text{Beta}(1, 1))}_{\text{evidential regularization}}. \tag{5}$$

where $\lambda > 0$ is a regularization weight (set to $10^{-3}$ in our experiments). The total loss is computed as a weighted sum across dimensions, with $w_A$, $w_B$, and $w_I$ all set to 1 in this paper. $\mathcal{L} = w_A \mathcal{L}_A + w_B \mathcal{L}_B + w_I \mathcal{L}_I$. Unlike unsupervised methods (e.g., Variational Autoencoders), our approach emphasizes interpretability, producing ABI values that are semantically meaningful and directly usable for trust-aware decision-making. Model details are provided in Appendix B.3.

**Online Update and Smoothing.** At inference time, the model produces $(\alpha_d, \beta_d)$ for each dimension, from which we compute the posterior mean and confidence. In addition to these instantaneous estimates, we maintain a smoothed posterior by treating the predicted mean $\mu_d$ as soft evidence:

$$\alpha_d^{(t)} \leftarrow \rho\,\alpha_d^{(t-1)} + \kappa\mu_d, \quad \beta_d^{(t)} \leftarrow \rho\,\beta_d^{(t-1)} + \kappa(1 - \mu_d), \tag{6}$$

where $\rho \in (0, 1)$ is a forgetting factor and $\kappa$ caps the evidence strength. In our implementation, we set $\rho = 0.999$ and define $\kappa = \min(S_{\text{model}}, 2.0)$, where $S_{\text{model}} = \alpha_d + \beta_d$ is the evidence strength predicted by the model. This smoothing stabilizes long-term estimates.

### 4.5 TRAINING: TRUST CO-PLAY

**Generating the Trustee Population.** Each trustee agent is trained with a base reward that combines *benevolence* and *integrity* components: $R_i^{\text{base}} = R_i^{(B)} + R_i^{(I)}$. The training objective of an ABI-grounded *trustee* agent is:

$$J(\pi_i) = \mathbb{E}_{\tau \sim \pi_i}\left[ \sum_t \left( R_i^{\text{base}}(s_t, a_t) \right) \right], \tag{7}$$

By varying the parameters in Eqs. 2 and 3, we generate different reward functions and thus obtain trustee agents with diverse benevolence-integrity profiles. To further diversify the population, we vary their *ability* by adjusting the rationality coefficient $\beta_i$ in the Boltzmann policy (Eq. 1). We trained each trustee agent using a pairing scheme, where it was paired with a complementary partner (e.g., a high-benevolence trustee that provides help was paired with a low-benevolence partner that receives help). Detailed implementation is provided in the Appendix B.2.

**Trust Co-Play.** With the trustee population established, we first train the ABI inference model, followed by the TRUSTPOMDP-based trustor. Using the same pairing scheme, we collect trajectories from trustee agents, each labeled with its ABI type, yielding training data $(\tau, \theta) \in \mathcal{T} \times \Theta$, where $\tau$ is a trajectory and $\theta$ the latent ABI label. These pairs are then used to train the inference model described in Sec. 4.3.

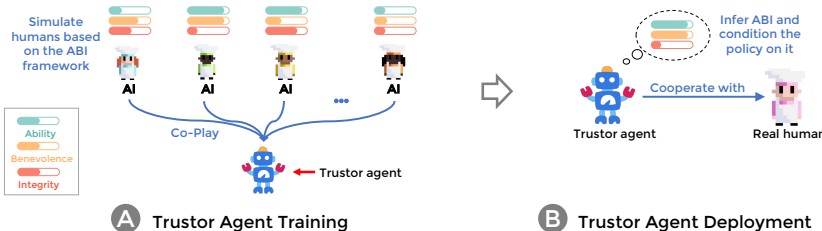

Figure 3: Illustration of Trust Co-Play. (a) The TRUSTPOMDP-based trustor agent is trained through co-play with a diverse set of trustee agents exhibiting varying levels of Ability, Benevolence, and Integrity. (b) The trained trustor agent can then collaborate with real humans, inferring their ABI and conditioning its policy accordingly.

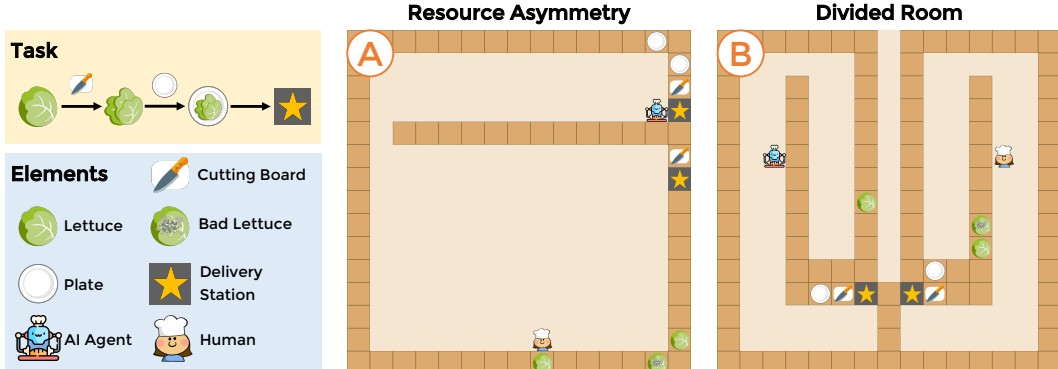

Figure 4: Task and two layouts in Overcooked. In this task, a human and an AI agent collaborate under time constraints to prepare and deliver as many lettuce salads as possible. We design two layouts—(A) *Resource Asymmetry* and (B) *Divided Room*—to induce trust-related challenges. In both, the human partner's ABI trait can be uncertainty. For instance, in (A), when the human carries a lettuce toward the bottom cutting board, the AI cannot tell whether the human intends to hand it over or plate it themselves after chopping. Such ambiguity creates a trust dilemma: the AI must decide whether to rely on the human, where misplaced trust can waste time or cause failure.

With the ABI inference model, finally, we train the trustor agent via co-play with the trustee population (Figure 3). In each episode, a trustee agent is sampled, and the inference model continuously updates the trustor's belief about the partner's traits, producing six signals $(A_\text{value}, A_\text{confidence}, B_\text{value}, B_\text{confidence}, I_\text{value}, I_\text{confidence})$. These signals are appended to the trustor's observations, enabling ABI-conditioned policy learning. The trustor is trained with Proximal Policy Optimization (PPO). Full model and training details are provided in Appendix B.3.

## 5 EXPERIMENT 1: EVALUATION WITH SIMULATED AGENTS

We evaluate our approach in Overcooked, a widely used testbed for studying human–AI cooperation (Carroll et al., 2019; Wang et al., 2024; Hong et al., 2023). Prior work in Overcooked has largely focused on coordination and collision avoidance, while overlooking trust as a key factor. Trust becomes critical under uncertainty (when a partner's trustworthiness is unknown) and risk (when misplaced trust leads to loss) (Mayer et al., 1995), yet standard Overcooked layouts rarely capture such dynamics. To evaluate our method in trust-sensitive settings, we designed new layouts where agents must decide whether to trust their partners under uncertainty. Misplaced trust in these layouts leads to negative consequences, such as wasted time and reduced scores.

## 5.1 METHOD

**Task and Environment.** In our setting, two agents must prepare and deliver as many lettuce salads as possible within a limited time. Each salad requires a sequence of actions: retrieving a lettuce, chopping it on a cutting board, fetching a plate, plating the salad, and delivering it (Figure 4). We first designed two trust-sensitive layouts (Figure 4): (1) **Resource Asymmetry**, where key resources lie on one side of the map, and (2) **Divided Room**, where agents operate in separate areas with asymmetric access. In both layouts, the trustee's intentions can sometimes be temporarily ambiguous (the *ambiguity zone*, described later), forcing the trustor to decide whether to wait for help or act independently. This can be a risky decision since misplaced trust (trusting an unreliable partner or distrusting a reliable one) can waste time and even cause task failure. To test generalizability, we also create easier variants of these layouts with rearranged item locations, called **Resource Asymmetry-Easy** and **Divided Room-Easy**, where the trustee agent's intention and trustworthiness are more perceptible (shown in Appendix C.1).

In Figure 4(a), the AI is positioned near the plates and the human near the lettuce. Ideally, the human would pass the lettuce, but this may be hindered by low ability (inefficient execution), low benevolence (withholding help), or low integrity (using bad lettuce). Detecting such traits is especially difficult in the *trait ambiguity zone*, where intentions and ABI remain unclear. For example, if the human moves right before picking up lettuce, their integrity is uncertain (will they use bad lettuce?), and if they carry lettuce toward the bottom cutting board, their benevolence is uncertain (will they pass it or keep it?). In these cases, the AI must decide whether to trust or act independently: misplaced trust wastes time, while misplaced distrust forfeits potential collaboration.

**Baselines and Evaluation** We compare our method with several baselines, including an ablated version of our model (*basic POMDP*), which is trained with the trustee agent population but does not infer or condition on ABI. We also evaluate against widely-recognized zero-shot coordination approaches such as Fictitious Co-Play (FCP) (Strouse et al., 2021) and Maximum Entropy Population-based training (MEP) (Zhao et al., 2023). Following prior work (Wang et al., 2024; Yu et al., 2023a), we construct a set of rule-based agents as deployment-time partners. We deliberately use rule-based behaviors—rather than learned agents—to create a clear distribution shift from the trustee population used during training, enabling a stronger test of the trustor agent's generalization. Implementation details are provided in the Appendix B.4.

## 5.2 RESULTS

We evaluate each method on four layouts with an episode length of $H = 400$ steps. For every *layout-method-partner* combination, we run 10 simulations and report the mean team reward per episode with 95% confidence intervals. We employed the Mann–Whitney U test with posthoc correction for the statistical analysis. As shown in Figure 5, TRUSTPOMDP achieves higher team rewards than other baselines ($p < 0.001$ for all). Trajectory analysis further reveals cases of *under-trust* and *over-trust* in baseline agents (Figure 7). For instance, when a benevolent partner (bottom) attempted to pass lettuce to the upper agent, FCP and MEP agents (upper) redundantly fetched lettuce independently, lowering efficiency. Conversely, when the partner was low in benevolence, the POMDP agent waited in vain, wasting valuable time. In contrast, the TRUSTPOMDP agent inferred the partner's benevolence from behavioral history and adapted its strategy accordingly. Detailed results are shown in Appendix C.3.

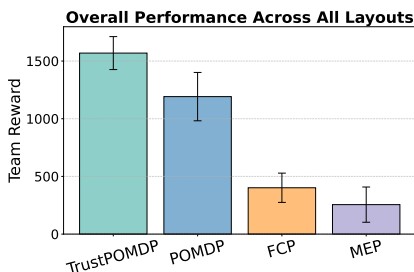

Figure 5: Overall team performance in Experiment 1 with simulated agents across four layouts, reported as means with 95% confidence intervals.

We also tested how well the ABI inference model can handle midway changes of the partner agent's behaviors, for example when a partner that is initially cooperative becomes uncooperative, or vice versa. Figure 6 show the dynamics of the inferred ABI in the *Divided Room* layout. Overall, when the partner's benevolence changes abruptly, the inferred belief is updated rapidly, although a delay is needed for updating. This demonstrates both the responsiveness of the ABI inference mechanism

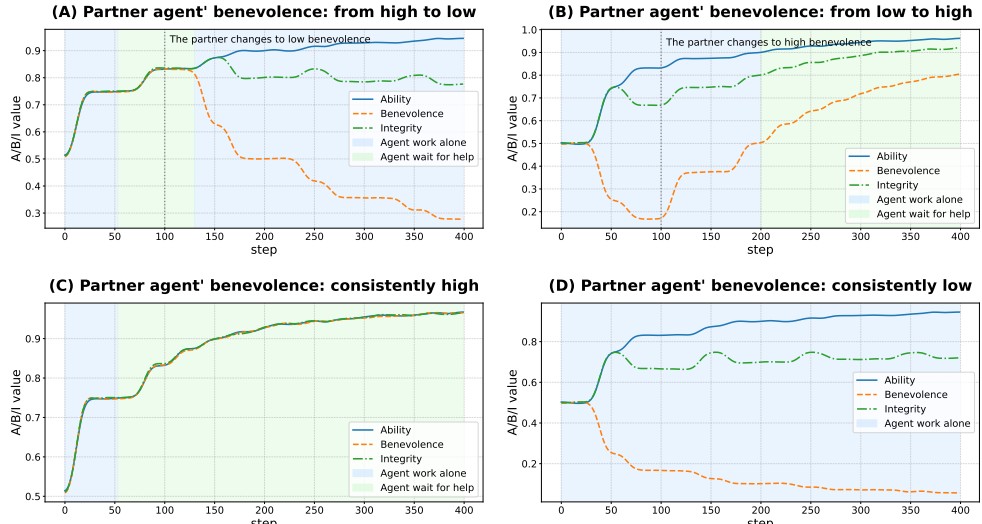

Figure 6: Dynamics of inferred ABI in the Divided Room layout, illustrated using Benevolence. The default TrustPOMDP policy is to work alone. (a) When the partner's benevolence switches from high to low at step 100, the agent initially waits for help as its belief increases, then returns to working alone once the partner moves away, even though inferred benevolence remains above 0.5, as environmental cues outweigh ABI under the default policy. (b) When benevolence changes from low to high, the agent adapts from working alone to waiting for help after the inferred benevolence exceeds 0.5. (c) With consistently high benevolence, the agent shifts from working alone to waiting for help after completing one solo dish. (d) With consistently low benevolence, the agent works alone throughout.

and the effectiveness of the ABI-adaptive TrustPOMDP policy in capturing and reacting to dynamic changes in partner behavior. We also provide another case in the *Resource Asymmetry* layout (see Figure 11 in the Appendix).

## 6 EXPERIMENT 2: EVALUATION WITH HUMAN PARTICIPANTS

### 6.1 METHOD

**Task and Participants.** We used the same task and environment as in Experiment 1. We recruited 106 participants from Prolific (59 female, 47 female; age = 36.25 ± 10.98).

**Experimental Design.** We compared the TRUSTPOMDP-based trustor agent with POMDP, FCP, and MEP agents. We employed a within-subjects design in which each participant collaborated with all four AI agents, with the order of agents counterbalanced.

**Experimental Procedure.** Each participant completed four tasks, interacting once with each AI agent in a counterbalanced order. Each task consisted of two rounds of 200 steps, resulting in a total of eight rounds per participant. For each participant, one layout was randomly selected from the four available layouts and used consistently across all four tasks, while the AI agent changed after each task. The AI agents were distinguished by color, but their underlying models were not revealed. Participants were explicitly informed that they were not required to play optimally and were free to interact with the AI in any way they preferred, allowing our method to be evaluated under a wide range of natural and diverse human strategies.

Before starting, participants were briefed on the study and provided informed consent. In the first task, at the beginning of each round, they specified the persona they wished to enact. In the subsequent tasks, they replayed the same personas to ensure comparability. After each task, participants completed a questionnaire assessing their collaborative experience and perceptions of the AI partner. Additional details about the experimental platform are provided in the Appendix C.8.

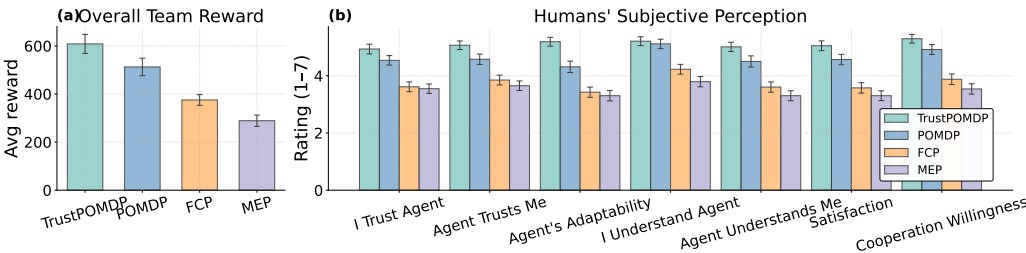

Figure 7: Qualitative observations in both Exp 1 and Exp 2. When the trustee agent (bottom) is benevolent, the TRUSTPOMDP agent learns to wait for assistance, enabling efficient collaboration. In contrast, the under-trusting agents (FCP and MEP) act independently, reducing efficiency. Conversely, when the trustee agent is not benevolent, TRUSTPOMDP adapts by working alone, whereas the over-trusting agent (POMDP) waits excessively, resulting in wasted time.

Figure 8: Overall team performance and participants' subjective perceptions in Experiment 2 across four layouts, shown with means and 95% confidence intervals.

## 6.2 RESULTS

We deployed the Friedman test with Holm posthoc correction for data analysis. Figure 8 summarizes the results. Overall, the TRUSTPOMDP agent significantly outperformed all baselines (vs. POMDP: $p < 0.01$, vs. FCP, MEP $p < 0.001$). Participants' subjective feedback echoed these findings. Participants reported greater trust in the TRUSTPOMDP agent, perceived its trust calibration as more appropriate, and rated it as more adaptable. They also thought the TrustPOMDP agent could better understand them. This fostered higher cooperation satisfaction and a stronger willingness to collaborate. Together, these results show that conditioning on inferred ABI enables more adaptive coordination and improves both performance and user experience, underscoring the value of equipping AI agents with human-like trust reasoning. Detailed statistical analysis and results each spacific layout are provided in Appendix C.7.

## 7 CONCLUSION

We have successfully demonstrated that equipping AI agents with human-like trust beliefs enhances their ability to cooperate with humans in the case where their competences and incentives are diverse. Our unique approach was to formulate a theory-informed and POMPD-compatible trust model that characterizes human partners along just three dimensions—ability, benevolence, and integrity, yet capturing a broad spectrum of human behaviors. Our evaluation shows that TRUSTPOMDPs adapt more effectively and achieve higher team performance than baseline agents when collaborating with human partners of varying trustworthiness. Participants also reported a better collaboration experience with our agent. Overall, these findings provide initial evidence that incorporating human-like trust mechanisms can substantially enhance cooperative AI.

REPRODUCIBILITY STATEMENT

We provide detailed implementation information for all models as well as the full description of the user study in the Appendix. In addition, the supplementary material includes our code, trained models, and the raw data from the user study.

ETHICS STATEMENT

This study included a user experiment conducted in accordance with local ethical requirements. We ensured that the experiment posed no harm to participants, informed them that they could withdraw at any time, and guaranteed that all data were collected anonymously and used solely for aggregate statistical analysis.

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

# A    CLAIMS AND PROOFS

Here, we present some intuitive but important claims and provide proofs.

## A.1    TRUSTPOMDP IS STILL A POMDP

**Proposition 1.** TRUSTPOMDP *preserves the Markov property and is a POMDP.*

*Proof.* Augment the state to $x_t = (s_t, z_t) \in \widetilde{\mathcal{S}} = \mathcal{S} \times \mathcal{Z}$. Human actions are drawn from $\pi_H(a_t^H \mid s_t, z_t)$, the physical transition from $T(s_{t+1} \mid s_t, a_t^{\mathrm{AI}}, a_t^H)$, and ABI dynamics from $\Xi(z_{t+1} \mid s_t, z_t, a_t^{\mathrm{AI}}, a_t^H, s_{t+1})$. Marginalizing $a_t^H$ gives the single-agent kernel

$$\widetilde{T}(x_{t+1} \mid x_t, a_t^{\mathrm{AI}}) = \sum_{a_t^H} T(s_{t+1} \mid s_t, a_t^{\mathrm{AI}}, a_t^H)\, \pi_H(a_t^H \mid s_t, z_t)\, \Xi(z_{t+1} \mid s_t, z_t, a_t^{\mathrm{AI}}, a_t^H, s_{t+1}).$$

Hence

$$\Pr(x_{t+1} \in A \mid x_{0:t}, a_{0:t}^{\mathrm{AI}}) = \Pr(x_{t+1} \in A \mid x_t, a_t^{\mathrm{AI}}) = \int_A \widetilde{T}(dx' \mid x_t, a_t^{\mathrm{AI}}),$$

so $\{x_t\}$ is Markov (under AI control). The AI's observation is $o_{t+1} \sim \widetilde{O}(\cdot \mid x_{t+1}, a_t^{\mathrm{AI}})$ with $\widetilde{O}(o \mid x', a) = O(o \mid s')$, and its one-step reward is $\widetilde{R}(x_t, a_t^{\mathrm{AI}}) = \mathbb{E}_{a_t^H \sim \pi_H(\cdot \mid s_t, z_t)}[R(s_t, a_t^{\mathrm{AI}}, a_t^H)]$. Therefore the control problem is the standard POMDP $\widetilde{\mathcal{M}} = \langle \widetilde{\mathcal{S}}, \mathcal{A}, \mathcal{O}, \widetilde{T}, \widetilde{O}, \widetilde{R}, \gamma \rangle$. Any statistic such as $\hat{z}_t = \mathcal{U}(h_t)$ is computed from observations and does not alter $(\widetilde{T}, \widetilde{O})$, hence does not affect Markovity. □

## A.2 ASSUMING ABI-LIKE PARTNERS, INFERRING ABI IS BENEFICIAL

**Notation and setup.**

- $\Theta = \{\theta_1, \ldots, \theta_N\}$: the set of latent ABI types of the human partner; $\theta \in \Theta$ is the true type, with prior $p_i = \Pr(\theta = \theta_i)$ and $\sum_i p_i = 1$.

- $a_t \in \mathcal{A}$: the AI's action at time $t$; $a', u$ denote generic actions.

- $r_t$: the immediate reward at time $t$; $\gamma \in (0, 1)$: the discount factor; $T$: the horizon (finite or infinite).

- $\mathcal{I}$: information available at time $t$ (e.g., observations and known model); $\mathcal{I}^+$: information after executing $a_t$ and transitioning to $t+1$.

- $Q(a \mid \mathcal{I})$: the *true* action-value under information $\mathcal{I}$ (with optimal continuation):

$$Q(a \mid \mathcal{I}) = \mathbb{E}\left[r_t + \gamma \max_{a'} Q(a' \mid \mathcal{I}^+) \,\Big|\, \mathcal{I}, \, a_t = a\right].$$

- **Base (ABI-agnostic) policy**: does not infer ABI; actions do not condition on $\theta$.

- **Trust-aware (ABI-inferencing) policy**: computes an ABI estimate $\hat{z}_t$ from available evidence (e.g., an inference module over observations) and allows actions to depend on $\hat{z}_t$.

**ABI-like (separability) assumption.** The partner is *ABI-like* if there exists a set of decision points with positive probability at which type-optimal actions differ across types; i.e., there exist $i \neq j$ and $a \neq a'$ such that

$$a \in \arg\max_u Q(u \mid \theta = \theta_i), \qquad a' \in \arg\max_u Q(u \mid \theta = \theta_j).$$

**Proposition 2.** *If the partner is ABI-like and the ABI estimate $\hat{z}_t$ is non-degenerate (it carries non-trivial information about $\theta$), then a trust-aware policy that conditions on $\hat{z}_t$ achieves a strictly higher expected discounted return than any base policy that does not infer ABI.*

*Proof.* Let $b_t(\theta) = \Pr(\theta \mid \text{current evidence})$ be the base policy's belief over types, and let $b_t^\sigma(\theta) = \Pr(\theta \mid \text{current evidence}, \hat{z}_t)$ be the belief after incorporating the ABI estimate. Define the respective one-step greedy actions:

$$a_t^{\text{base}} \in \arg\max_a \mathbb{E}_{\theta \sim b_t}\big[Q(a \mid \theta)\big], \qquad a_t^{\text{trust}} \in \arg\max_a \mathbb{E}_{\theta \sim b_t^\sigma}\big[Q(a \mid \theta)\big].$$

Define the instantaneous gain

$$\Delta_t := \mathbb{E}_{\theta \sim b_t^\sigma}\big[Q(a_t^{\text{trust}} \mid \theta)\big] - \mathbb{E}_{\theta \sim b_t}\big[Q(a_t^{\text{base}} \mid \theta)\big].$$

By optimality, $\Delta_t \geq 0$. Under the ABI-like assumption, there is a set of positive probability on which the $Q$-maximizing action depends on $\theta$; since $\hat{z}_t$ is non-degenerate, with positive probability the updated belief $b_t^\sigma$ shifts toward the realized type enough to change the greedy action and strictly increase the inner expectation, hence $\Pr(\Delta_t > 0) > 0$. Therefore,

$$\mathbb{E}\left[\sum_{t=0}^T \gamma^t \Delta_t\right] > 0,$$

which implies that the trust-aware policy attains a strictly higher expected discounted return than the base policy. □

## A.3 THE BENEFIT OF ABI ESPECIALLY COMES FROM BETTER DISAMBIGUATION IN TRAIT-AMBIGUITY ZONES

**Definition (Trait-ambiguity zone).** A *trait-ambiguity zone* is any set $\mathcal{U}$ of AI-observable observations (or observation sequences) such that, for all types $i, j$ in $\Theta$,

$$p(\mathbf{o} \mid \theta_i) = p(\mathbf{o} \mid \theta_j), \qquad p(\mathbf{s}' \mid \mathbf{s}, a, \theta_i) = p(\mathbf{s}' \mid \mathbf{s}, a, \theta_j) \quad (\forall \mathbf{o} \in \mathcal{U}, \ \forall a),$$

so conditioning on $\mathcal{U}$ does not update the posterior over $\theta$ (posterior = prior).

**Proposition 3.** *In trait-ambiguity zones (observations look the same across ABI types), any ABI-nonadaptive policy can only choose a single, average-optimal action. If the human is ABI-like (type-separable payoffs) and ABI inference is above chance, then an ABI-adaptive policy that conditions on the inferred type strictly outperforms all ABI-nonadaptive policies in such zones.*

*Proof.* **Setup.** Let partner's trait type $\theta \in \Theta = \{\theta_1, \ldots, \theta_N\}$ with prior $p_i = \Pr(\theta = \theta_i)$. At decision epoch $t^*$ (discount $\gamma \in (0, 1)$), choosing $a \in \{1, \ldots, N\}$ yields payoff $R_{a,i}$ if the true type is $\theta_i$ (later rewards are zero), so the discounted return is $\gamma^{t^*} R_{a,i}$. Define the prior-weighted value of any *fixed* action and its best value:

$$U_a := \sum_{i=1}^N p_i R_{a,i}, \qquad B^\star := \max_a U_a.$$

In a trait-ambiguity zone, an ABI-nonadaptive (observation-only) policy must commit to a single $a$, achieving at most

$$V_{\text{non}}^\star = \gamma^{t^*} B^\star.$$

An ABI-adaptive policy first infers $\hat{\theta} \in \Theta$ with confusion probabilities $P_{j|i} := \Pr(\hat{\theta} = \theta_j \mid \theta = \theta_i)$ and then plays $a = \hat{\theta}$, achieving

$$V_{\text{adapt}} = \gamma^{t^*} \sum_{i=1}^N p_i \sum_{j=1}^N P_{j|i} R_{j,i}.$$

**Gap formula.** Subtracting the nonadaptive bound gives the exact decomposition

$$V_{\text{adapt}} - \gamma^{t^*} B^\star = \gamma^{t^*} \left( \sum_{i=1}^N p_i \sum_{j=1}^N P_{j|i} R_{j,i} - \max_a \sum_{i=1}^N p_i R_{a,i} \right). \tag{8}$$

**Sufficient condition.** Assume *ABI-like separability*: for each type $i$, the type-matched action strictly dominates all others,

$$\Delta_i := R_{i,i} - \max_{a \neq i} R_{a,i} > 0.$$

Let the *accuracy margin* on column $i$ be

$$\varepsilon_i := P_{i|i} - \max_{a \neq i} P_{a|i}.$$

If there exists a subset $\mathcal{I} \subseteq \{1, \ldots, N\}$ with positive prior mass $\sum_{i \in \mathcal{I}} p_i > 0$ such that $\varepsilon_i > 0$ for all $i \in \mathcal{I}$ (i.e., inference is above chance on those types), then a standard column-wise comparison yields

$$\sum_{i=1}^N p_i \sum_{j=1}^N P_{j|i} R_{j,i} - \max_a \sum_{i=1}^N p_i R_{a,i} \geq \sum_{i \in \mathcal{I}} p_i \varepsilon_i \Delta_i > 0.$$

Plugging this lower bound into equation 8 gives $V_{\text{adapt}} > \gamma^{t^*} B^\star$.

**Intuition.** In trait-ambiguity zones, observation-only policies are forced to make pooled (average) decisions. ABI adaptation converts pooled decisions into type-contingent ones. Whenever the inference is even modestly better than chance on a nontrivial set of types, the positive margins $\varepsilon_i$ combine with the type-separation gaps $\Delta_i$ to produce a strictly positive improvement. $\square$

| Layout | Trustee agent | Paired trustor agent | Reward shaping |
|---|---|---|---|
| Resource Asymmetry | highB_highI (1) | lowB_highI | trustor pick up lettuce from counter + 50 (first time only)
trustee go to cutting board + 50 (first time only)
trustee pass chopped lettuce + 50 (first time only)
trustor wait for help + 50 (first time only) |
|  | highB_highI (2) | lowB_highI | trustor pick up lettuce from counter + 50 (first time only)
trustee get lettuce + 50 (first time only)
trustee pass lettuce + 50 (first time only)
trustor wait for help + 50 (first time only) |
|  | highB_lowI | lowB_lowI | trustor pick up bad lettuce from counter + 50 (first time only)
trustee get bad lettuce + 100 (first time only)
trustee pass bad lettuce + 50 (first time only)
trustor wait for help + 50 (first time only)
trustor/trustee plate bad lettuce + 20 |
|  | lowB_highI | lowB_highI | - |
|  | lowB_lowI | lowB_highI | trustor plate bad lettuce - 200
trustor pick up bad lettuce - 100
trustee plate bad lettuce + 200
trustee pick up bad lettuce + 200 (first time only) |
| Resource Asymmetry easy | highB_highI (1) | lowB_highI | trustor pick up lettuce from counter + 50 (first time only)
trustee go to cutting board + 100 (first time only)
trustee pass chopped lettuce + 100 (first time only)
trustor wait for help + 50 (first time only) |
|  | highB_highI (2) | lowB_highI | trustor pick up lettuce from counter + 50 (first time only)
trustee get lettuce + 50 (first time only)
trustee pass lettuce + 50 (first time only)
trustor wait for help + 50 (first time only) |
|  | highB_lowI | lowB_lowI | trustor pick up bad lettuce from counter + 20 (first time only)
trustee get bad lettuce + 20 (first time only)
trustee pass bad lettuce + 20 (first time only)
trustor wait for help + 20 (first time only)
trusee plate bad lettuce + 20
trutor plate bad lettuce + 20 |
|  | lowB_highI | lowB_highI | - |
|  | lowB_lowI | lowB_highI | trustor plate bad lettuce - 200
trustor pick up bad lettuce - 100
trustee plate bad lettuce + 200
trustee pick up bad lettuce + 200 (first time only) |
| Divided Room | highB_highI (1) | lowB_highI | trustor pick up lettuce from counter + 50 (first time only)
trustee go to cutting board + 50 (first time only)
trustee pass chopped lettuce + 50 (first time only)
trustor wait for help + 500 (first time only) |
|  | highB_highI (2) | lowB_highI | trustor pick up lettuce from counter + 50 (first time only)
trustee get lettuce + 50 (first time only)
trustee pass lettuce + 50 (first time only)
trustor wait for help + 500 (first time only) |
|  | highB_lowI | lowB_lowI | trustor pick up bad lettuce from counter + 20 (first time only)
trustee pass bad lettuce + 20 (first time only)
trustor wait for help + 1000 (first time only)
trustor/trustee plate bad lettuce + 20 |
|  | lowB_highI | lowB_highI | - |
|  | lowB_lowI | lowB_highI | trustee plate bad lettuce + 20
trustee pick up bad lettuce + 200 (first time only) |
| Divided Room easy | highB_highI (1) | lowB_highI | trustor pick up lettuce from counter + 20 (first time only)
trustee go to cutting board + 50 (first time only)
trustee pass chopped lettuce + 50 (first time only)
trustor wait for help + 1000 (first time only)
trustor/trustee plate bad lettuce - 20
trustor/trustee pick up bad lettuce - 10 |
|  | highB_highI (2) | lowB_highI | trustor pick up lettuce from counter + 50 (first time only)
trustee get lettuce + 50 (first time only)
trustee pass lettuce + 50 (first time only)
trustor wait for help + 1000 (first time only)
trustor/trustee plate bad lettuce - 20
trustor/trustee pick up bad lettuce - 10 |
|  | highB_lowI | lowB_lowI | trustor pick up bad lettuce from counter + 20 (first time only)
trustee pass bad lettuce + 50 (first time only)
trustee [ass bad lettuce + 20 (first time only)
trustor wait for help + 50 (first time only)
trustor/trustee plate bad lettuce + 20 |
|  | lowB_highI | lowB_highI | trustor/trustee plate bad lettuce - 20
trustor/trustee pick bad lettuce - 10 |
|  | lowB_lowI | lowB_highI | trustee pick up bad lettuce + 20
trustee plate bad lettuce + 200
trustor pick up bad lettuce - 100
trustor plate bad lettuce -200 |

Table 1: Reward shaping used to derive different trustee agents.

# B    IMPLEMENTATION DETAILS

## B.1    ENVIRONMENT

**Observation.**    Each observation is represented as a 32-dimensional feature vector, consisting of: (1) the ego agent's absolute position and a binary flag indicating whether it is holding an object; (2) the relative position and holding status of its partner; (3) the relative positions and current states of all items in the environment with respect to the ego agent (e.g., whether a lettuce is chopped, or a plate/cutting board is occupied); and (4) a binary flag indicating which agent is the ego.

**Reward.**    The reward function is defined as follows:

- Cutting a lettuce: **+10**
- Plating a chopped lettuce: **+20**
- Delivering a correct dish: **+200**
- Delivering an incorrect item (e.g., an empty plate, a dish not on the menu): **–50**
- Each step taken: **–1**

**Action Space.**    The action space includes high-level discrete actions: "stay", "get lettuce", "get plate", "go to knife", "deliver", "chop", and "go to counter". These are supported by primitive actions: "left", "right", "up", and "down". High-level actions are executed via A* path planning to generate corresponding low-level movement sequences.

We choose high-level action abstraction over purely primitive actions for two reasons. First, it enhances sample efficiency and accelerates learning, especially in larger maps—crucial for our focus on trust dynamics rather than motor control. Second, high-level actions better reflect human reasoning patterns. For example, humans tend to think in terms of "getting lettuce" rather than low-level movements like "up-up-left". This abstraction enables agent behaviors that are more interpretable and trust-relevant.

## B.2    TRUSTEE AGENT

For each map, we constructed ten trustee agents with different ABI profiles: (1) highA–highB–highI–1, (2) highA–highB–highI–2, (3) highA–highB–lowI, (4) highA–lowB–highI, (5) highA–lowB–lowI, (6) lowA–highB–highI–1, (7) lowA–highB–highI–2, (8) lowA–highB–lowI, (9) lowA–lowB–highI, (10) lowA–lowB–lowI.

**Modeling ABI.** Table 2 summarizes the original ABI definitions in (Mayer et al., 1995) and our corresponding operationalizations.

| Dimension | Original Definition | Operationalization in This Work |
|---|---|---|
| **Ability** | The belief that the trustee has the group of skills, competencies, and characteristics that enable them to have influence within some specific domain (Mayer et al., 1995). | Operationalized as the agent's tendency to select the action with the highest expected reward in a given state. A more deterministic, goal-directed policy reflects higher ability. |
| **Benevolence** | The belief that the trustee will want to do good to the trustor, aside from an egocentric profit motive (Mayer et al., 1995). | Operationalized as the degree to which an agent values team success over personal gain. A more benevolent agent contributes to its partner's reward more heavily. |
| **Integrity** | The belief that a trustee adheres to a set of principles that the trustor finds acceptable (Mayer et al., 1995). | Operationalized as the agent's adherence to implicit norms or task constraints, such as avoiding shortcuts or unethical actions, even at the cost of immediate reward. |

Table 2: Original definitions of ABI dimensions (Mayer et al., 1995) and their operationalization in our framework.

For **Ability**, we adjust the parameter $\beta_i$ in Eq. 1. High-ability agents are modeled without Boltzmann sampling, equivalent to $\beta_i = +\infty$. Low-ability agents are modeled with $\beta_i = 0.3$, introducing stochasticity into their policies.

| Trustee Agent | Paired Partner Agent |
|---|---|
| High B, High I | Low B, High I |
| High B, Low I | Low B, Low I |
| Low B, High I | Low B, High I |
| Low B, Low I | Low B, High I |

Table 3: Pairings between trustee and partner agents. Trustee agents form the final population, while partner agents are used only for training.

For **Benevolence**, we define a set of credit-earning events incorporated into the reward function, such as chopping a vegetable (+10), plating (+20), and delivering a correct dish (+200). We then adjust the weighting parameter $\alpha$ in Eq. 2. In the high-benevolence condition, we set $\alpha = 0$, making the agent's reward fully determined by its partner's reward (and shared reward). In the low-benevolence condition, we set $\alpha = 1$, making the agent's reward fully self-centered.

For **Integrity**, we design norm-violating actions, such as using spoiled lettuce to prepare a dish. In the high-integrity condition, we set the parameter $\delta$ in Eq. 3 to zero or a negative value (depending on the layout). In the low-integrity condition, $\delta$ is set to a positive value, incentivizing norm-violating behavior.

We designed a agent-pairing scheme where each trustee agent is paired with a trustor partner (Table 3), rather than relying on the self-play approach. This explicit role assignment was intentional: self-play makes it difficult to establish clear distinctions between trustor and trustee, and often leads to coordination failures. For example, two high-benevolence agents may both attempt to help each other, resulting in ambiguous and unstable behaviors. Note that these trustor partners are only used for traingin trustee agents but not used for later stage.

Finally, to encourage trustee agents to better learn the intended behaviors, we incorporate additional reward shaping (Table A.3). For example, two versions of highA-highB-highI are derived based on different reward shaping for diversity.

**RL Algorithm and Hyperparameters.** We use Proximal Policy Optimization (PPO) for training. The model is trained with a learning rate of $3 \times 10^{-4}$, rollout horizon of 256 steps, and batch size of 128. Each update consists of 10 epochs of gradient descent. We use a discount factor of $\gamma = 0.95$ and GAE parameter $\lambda = 0.95$. The clipping range is set to 0.3, the entropy coefficient to 0.02 (except for the Divided Room layout, which is set to 0.05), and the value loss coefficient to 0.5. Gradients are clipped at 0.5. The policy and value networks are implemented as separate multilayer perceptrons with hidden layers of size 256, 128, and 64.

For each trustee agent, we trained $4.1 \times 10^6$ steps and ensured convergence.

### B.3 TRUSTPOMDP-BASED TRUSTOR AGENT

**Reward.** The reward function for the TRUSTPOMDP-based trustor agent is identical to the team reward, without any additional modification or reward shaping.

**Hyperparameters.** To accelerate training, we use 8 parallel environments for rollout collection, set $n\_steps = 3600$ and batch size to 600, while keeping all other hyperparameters the same as those used for the trustee agents.

**ABI Inference Model** Each state $x_t \in \mathbb{R}^D$ is first linearly projected into a hidden space of dimension $H = 32$ and encoded by a lightweight Transformer encoder (1 layer, 2 attention heads, feed-forward size $2H = 64$, ReLU activation, batch-first). This produces contextualized representations $h_{1:T}$.

For each trust dimension $d \in \{A, B, I\}$, we construct a dimension-specific temporal mask $M_d$ that retains only the most recent $k_d$ steps ($k_A = 15$, $k_B = 30$, $k_I = 30$ for Resource Asymmetry, Resource Asymmetry-Easy, Divided Room-Easy, $k_A = 10$, $k_B = 10$, $k_I = 10$ for Divided Room), combined with padding masks for variable sequence lengths. A shared learnable attention vector

$v \in \mathbb{R}^H$ is then used to compute an attention-pooled summary:

$$\tilde{h}_d = \sum_{t=1}^{T} w_{t,d} h_t, \quad w_{t,d} = \frac{\exp(h_t^\top v)}{\sum_{j \in M_d} \exp(h_j^\top v)},$$

where masked positions are excluded.

The pooled representation $\tilde{h}_d$ is passed through a dimension-specific MLP head (Linear($H \rightarrow 64$) + ReLU), followed by two linear layers that output the Beta distribution parameters:

$$\alpha_d = \text{softplus}(f_d^\alpha(\tilde{h}_d)) + \epsilon, \quad \beta_d = \text{softplus}(f_d^\beta(\tilde{h}_d)) + \epsilon,$$

with $\epsilon = 10^{-4}$ ensuring numerical stability and $\alpha_d, \beta_d > 0$. The Beta mean $p_d = \alpha_d/(\alpha_d + \beta_d)$ represents the inferred trust value, while the strength $S_d = \alpha_d + \beta_d$ captures the model's certainty.

The model parameters are optimized with Adam (learning rate $1 \times 10^{-3}$). A simpler baseline variant replaces the Beta outputs with sigmoid predictions for each trust dimension, while using the same encoder and attention-pooling backbone.

To improve sampling efficiency, we collect a trajectory snapshot whenever the trustee agent places down an item (of any type). The same event is used during deployment, where the trustor agent updates its ABI inference in real time whenever the trustee agent puts down an item. In addition, the historical observations used for inference include only the partner agent's position and the item being held (a 6-dimensional vector), rather than the full observation. This design prevents the trustor agent's own behavior from influencing the inference of the trustee agent's ABI.

**Conditioning the policy on ABI.** We append a six-dimensional ABI context to each observation, $(A_{\text{value}}, B_{\text{value}}, I_{\text{value}}, A_{\text{confidence}}, B_{\text{confidence}}, I_{\text{confidence}})$, where $A_{\text{value}}, B_{\text{value}}, I_{\text{value}} \in [0, 1]$ are the inferred ABI values and $A_{\text{confidence}}, B_{\text{confidence}}, I_{\text{confidence}} \in [0, 1]$ are confidences. The extractor *ABI-GatedExtractorWithConf* splits the input into the non-ABI part $x$ and the ABI context. The non-ABI features are encoded by a shared backbone $f = \phi(x) \in \mathbb{R}^D$ (two-layer MLP with ReLU).

To allow the policy to react more differently to high vs. low ABI, we utilize signed gates

$$A^+ = \text{ReLU}(binary(A)), \quad A^- = \text{ReLU}(-binary(A)),$$

(and analogously for $B, I$). First, we binarize A, B, I based on a threshold 0.5, then we use ReLU activation function to process the binary ABI values to form a gate. Each gate modulates the shared feature $f$, yielding six gated streams ($f \odot A^+$, $f \odot A^-$, $f \odot B^+$, $f \odot B^-$, $f \odot I^+$, $f \odot I^-$). These are concatenated with the raw ABI signals and confidences:

$$\text{feat} = \left[ f \odot A^+; f \odot A^-; f \odot B^+; f \odot B^-; f \odot I^+; f \odot I^-; A, B, I, \text{conf}_A, \text{conf}_B, \text{conf}_I \right],$$

resulting in a feature vector of dimension $6 \cdot base\_dim + 6$ (with $base\_dim = 64$ by default). The actor–critic heads then operate on this ABI-aware representation. Concretely, we use Stable-Baselines3 with a custom feature extractor (*ABIGatedExtractorWithConf*) and set the base hidden dimension to 64. The policy and value networks (*pi* and *vf*) are both two-layer MLPs with sizes [128, 64]. Thus, both the policy $\pi$ and value function $V$ are conditioned on features that (i) separate positive and negative evidence per ABI dimension, (ii) scale their influence by certainty, and (iii) retain the raw ABI and confidence values, enabling the agent to adapt to the inferred partner profile.

**Training.** Our goal is to construct a *trust-critical* environment whose core feature is the presence of an *ambiguity zone*. When the partner operates within this zone, the trustor agent must make an accurate trust judgment; otherwise, it will incur time loss and may lead to coordination failure. To increase both the proportion of ambiguity zones within an episode and the salience of the consequences of trust and mistrust, we periodically reset the positions of key environmental elements. Specifically, positions are reset every 100 steps (and every 70 steps in the Resource Asymmetry layout), forcing the agents to repeatedly re-enter ambiguity zones. Each episode consists of 400 steps.

The agent was trained for $8 \times 10^6$ updates, which, with 8 parallel environments, corresponds to a total of $6.4 \times 10^7$ environment steps.

### B.4 BASELINES

**FCP.** Fictitious Co-Play (FCP) is a two-stage training framework. In the first stage, it builds a diverse partner population by pre-training self-play (SP) agents with different random seeds and saving multiple checkpoints at different training stages to capture policies of varying "capabilities." In the second stage, an FCP agent is trained by repeatedly playing against partners sampled from this population. In our implementation, we trained five SP agents with seeds 15, 25, 35, 45, and 55, each for 6.1M steps. For each SP agent, we saved checkpoints at steps 100k, 200k, 400k, 2M, and 6.1M, covering the full spectrum from early learning to convergence. This yields a partner population of $5 \times 5 = 25$ agents. In the second stage, we trained the FCP agent for $2 \times 10^7$ steps. The policy network architecture and hyperparameters for both SP and FCP agents match those used for the trustee agents described earlier.

**MEP.** Maximum Entropy Population-based training (MEP) is a variant of FCP. It introduces a maximum-entropy diversity bonus into the task reward, which encourages the population in the first stage to explore a wider range of strategies. In the second stage, a robust agent is trained by *rank-based prioritized sampling* from this population. Given the evaluation returns of the population, we rank partners by difficulty (lower return $\Rightarrow$ higher difficulty) and sample partners with probability proportional to $\text{rank}^\beta$. Here, $\beta$ controls the sharpness of the sampling distribution: $\beta = 0$ yields uniform sampling, $\beta = 1$ samples proportionally to rank, and larger $\beta$ further concentrates training on the most challenging partners. In our implementation, we constructed five SP agents with seeds 15, 25, 35, 45, and 55, using $\alpha = 1.0$ for the entropy bonus in the first stage, and $\beta = 3$ for prioritized sampling in the second stage following the original paper's setting. We trained the FCP agent for $2 \times 10^7$ steps. The policy network architecture and hyperparameters for both SP and MEP agents match those used for the trustee agents described earlier.

**POMDP.** The POMDP baseline uses exactly the same partner population as TrustPOMDP and is trained with identical RL hyperparameters. The POMDP agent was trained for $8 \times 10^6$ updates, which, with 8 parallel environments, corresponds to a total of $6.4 \times 10^7$ environment steps.

## C EXPERIMENT DETAILS

### C.1 ADDITIONAL LAYOUTS

In addition to the two layouts presented in Figure 4 (Resource Asymmetry and Divided Room), we also designed two simplified variants: Resource Asymmetry–Easy and Divided Room–Easy (Figure 9). The key difference is that the original layouts contain large ambiguity zones, where it is difficult to infer the trustee agent's intention from observation alone. By contrast, the Easy variants have little or no ambiguity. For example, in Figure 9(a), when the trustee agent on the right moves left, it is immediately clear that it intends to use the lettuce, while moving down-right reveals an intention to use the bad lettuce—making integrity easy to infer. Similarly, after picking up a vegetable, moving left indicates a willingness to help, while moving right implies self-serving behavior. After chopping, moving left suggests cooperation, whereas moving up suggests acting alone to complete the dish. The same logic applies to Figure 9(b). We introduced these Easy layouts primarily to examine under what conditions ABI inference provides meaningful benefits.

### C.2 RULE-BASED AGENTS IN EXPERIMENT 1

We designed nine rule-based agents, each focusing on a single type of behavior: *pass plate*, *pass lettuce*, *pass chopped lettuce*, *pass plated lettuce*, *pass dirty lettuce*, *pass chopped dirty lettuce*, *pass plated dirty lettuce*, *make clean salad alone*, and *make dirty salad alone*.

We observed that several of these agents, such as *pass lettuce*, *pass chopped lettuce*, *pass dirty lettuce*, *make clean salad alone*, and *make dirty salad alone*, exhibit behaviors similar to those in our trustee population. However, others—such as *pass plate*, *pass plated lettuce*, *pass chopped dirty lettuce*, and *pass plated dirty lettuce*—differ substantially from our trustee agents. This ensures a broader out-of-distribution (OOD) test set, providing a stronger evaluation of model generalization.

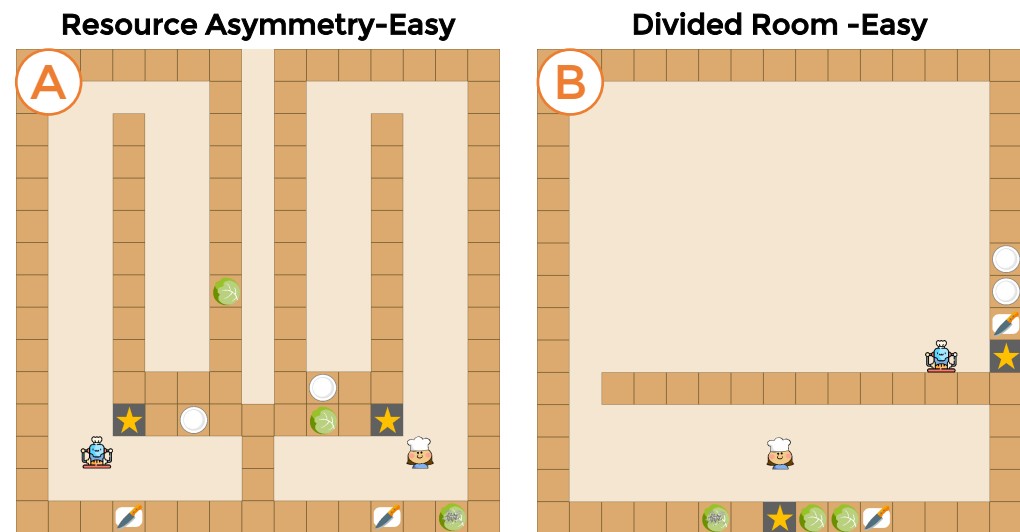

Figure 9: Another two layouts where the trustee agent's (the bottom and right one) intention is easier to perceive. In other words, the ambiguity zone is small.

During testing, we additionally duplicated the *pass lettuce* and *pass chopped lettuce* agents to balance the proportion of trustworthy and untrustworthy partners at approximately 1:1.

## C.3 ADDITIONAL RESULTS IN EXPERIMENT 1

Figure 10 shows the average team reward of the four models across the four layouts. On the Resource Asymmetry-Easy layout, the performance difference between TrustPOMDP and POMDP is negligible. We attribute this to the fact that POMDP agents also learned to trust their partners by default and therefore tend to wait for help initially. Given that the ambiguity zone in this layout is very small, the agent can quickly infer its partner's intention after a short observation period.

In contrast, on the Divided Room-Easy layout, although the ambiguity zone is similarly small, the POMDP agent learned to distrust its partner by default. As a result, it fails to exploit potential help from the partner and misses opportunities for cooperation. We argue that temporarily waiting for help is an effective strategy for probing and clarifying the partner's intention.

These findings further suggest that in low-ambiguity environments, if an agent adopts a strategy of briefly waiting to confirm the partner's intention, reasonable performance can be achieved even without explicit ABI inference. In such cases, training only with our constructed trustee agent population is sufficient. However, when the partner's traits exhibit higher ambiguity, ABI inference and policy conditioning become essential for achieving effective cooperation.

## C.4 ABI DYNAMICS IN RESOURCE ASYMMETRY LAYOUT

We test whether the ABI inference model can handle midway behavior changes of the partner agent. In this layout, we manipulate the partner agent's benevolence, including four situations: "consistently low", "consistently high", "from low to high", "from high to low". We visualize the temporal dynamics of ABI inference and corresponding behavioral adaptation of the TrustPOMDP agent (see Figure 11. The results show that our ABI inference model can effectively detect these mid-task changes, and the TrustPOMDP agent can adjust its behavior in response to the updated ABI belief.

## C.5 COMPARISON BETWEEN BINARY ABI AND CONTINUOUS ABI

We compared the performance between binary ABI-based TrustPOMDP and continuous ABI-based TrustPOMDP. Results are shown in Figure 12.

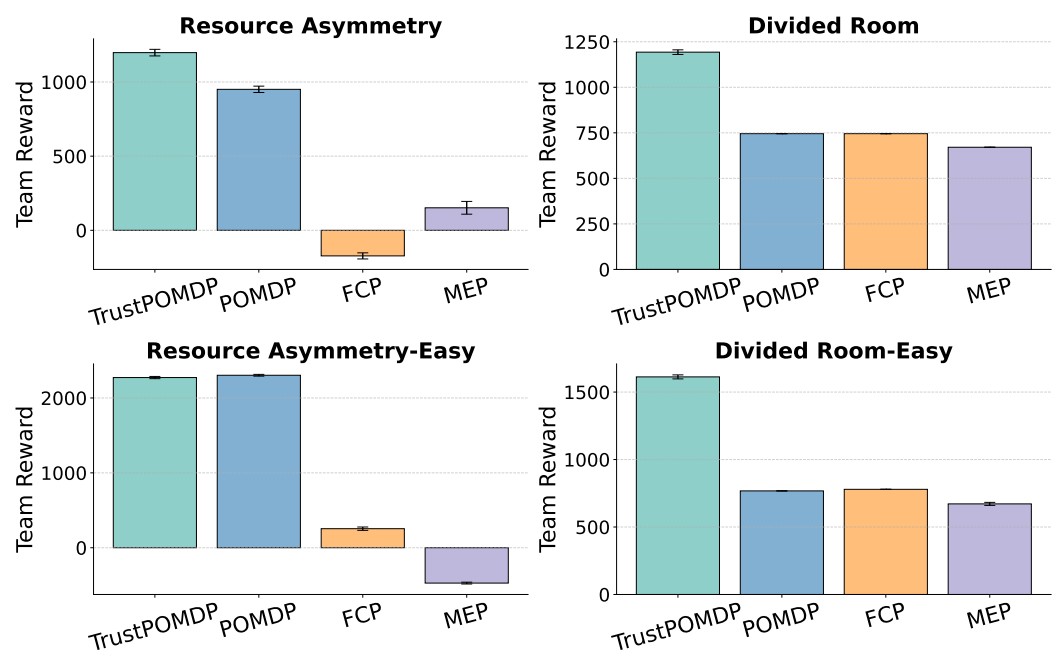

Figure 10: Detailed team performance in the four layouts.

## C.6 ANALYSIS OF WHY FCP AND MEP PERFORM BADLY

The simulation experiment results reveal a low performance of FCP and MEP agents. We provide the analysis of the reason.

The primary reason for the performance gap lies in the nature of the partner populations on which FCP and MEP rely. Both methods construct their partner populations using SP-trained agents. Due to decentralized training, SP agents naturally learn to solve tasks independently and tend to avoid behaviors that require active cooperation, resulting in consistently low Benevolence. In contrast, real humans often show a natural willingness to help teammates, and our ABI-producing agents explicitly cover systematic variations along the Benevolence dimension. Similarly, SP agents are trained with team reward objectives and therefore rarely learn behaviors corresponding to low Integrity (e.g., using dirty or suboptimal ingredients). In real interactions, however, humans may occasionally act in ways that violate optimal or normative behavior, whether intentionally or unintentionally. Our ABI-producing agents explicitly incorporate variability along the Integrity dimension, thereby capturing a broader and more realistic spectrum of partner behaviors. In summary, the SP-based populations used by FCP and MEP exhibit limited diversity, primarily clustering around low Benevolence and high Integrity. Although FCP and MEP introduce diversity along the Ability dimension by sampling agents from different training checkpoints, they still fail to capture the complex variations in mixed-motive or hidden-utility scenarios involving Benevolence and Integrity.

As a result, FCP- and MEP-based agents exhibit a systematic tendency to under-trust their partners. Even when a teammate shows willingness to assist, these agents still prefer to act independently, unless the help is made explicit, such as when the partner has already passed over a lettuce.

A secondary reason is the nature of our simulation study, which deliberately includes a set of rule-based and out-of-distribution (OOD) partner behaviors, such as deliberately handing over chopped bad vegetables. These behaviors are rare and irrational within the designed environment and are intended as stress tests to test model robustness. As a result, FCP and MEP perform particularly poorly in some layouts under these extreme conditions. For example, in the Resource Asymmetry layout, both agents can access global items, leading to stronger interference; when confronted with OOD partner behaviors, FCP- and MEP-based agents struggle more noticeably. In contrast, in the Divided Room layout, where the agents operate in relatively independent local environments, FCP- and MEP-based agents are still able to execute reasonably effective actions even when the partner

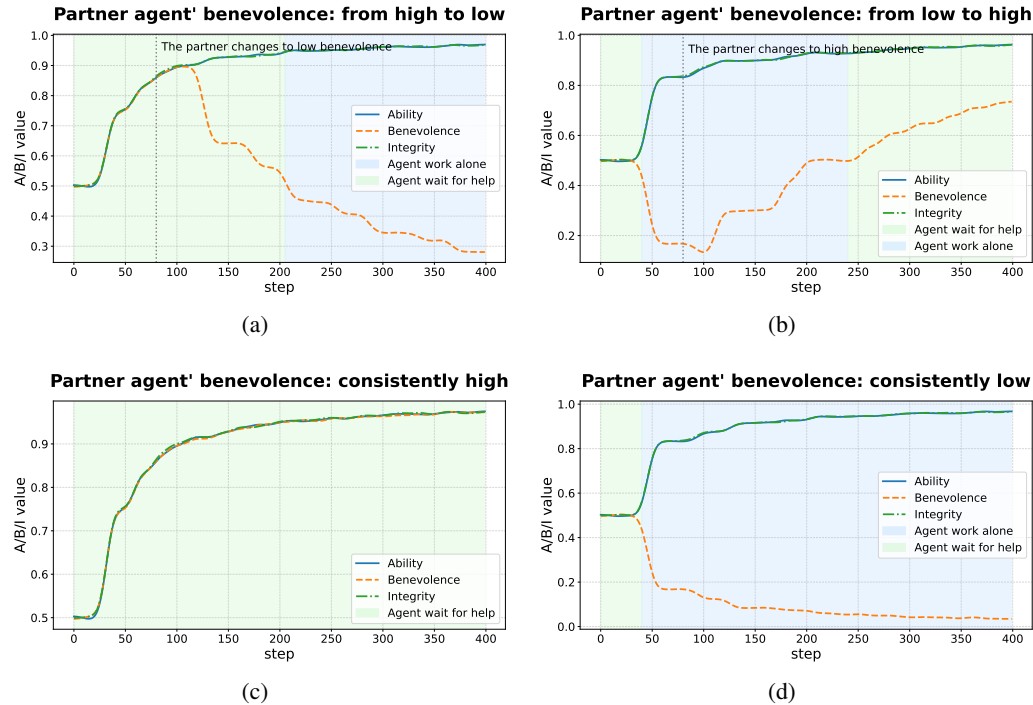

(a)

(b)

(c)

(d)

Figure 11: Dynamics of the inferred ABI in the Resource Asymmetry layout. We use Benevolence midway changes for illustration. In this layout, the default policy learned by TrustPOMDP is to wait for help. In (a), the partner's benevolence is initially high (e.g., passing lettuce) and then switches to low at step 80 (i.e., completing dishes alone). The TrustPOMDP agent first waits for the partner's help, then adapts to working alone as its benevolence belief decreases (starting from step 205, once the inferred benevolence is below 0.5). In (b), the partner's benevolence is initially low and then switches to high at step 80. Accordingly, the TrustPOMDP agent first waits for the partner's help and then transitions to working alone (starting from step 40). As the inferred benevolence gets updated, the TrustPOMDP agent later turns to wait for the partner's help again (starting from step 240). Note that there is a delay of belief updating as the TrustPOMDP agent needs to accumulate enough evidence. In (c), the partner consistently exhibits high benevolence, and the TrustPOMDP agent consistently waits for help throughout the interaction. In (d), the partner consistently exhibits low benevolence. Although the default policy is to wait for help, the TrustPOMDP agent initially waits for help and then adapts to working alone (starting from step 40). The inferred ABI values displayed in this figure are smoothed.

behaves in an OOD manner. Importantly, in our human study, where participants exhibit more realistic and less extreme behaviors, the performance of FCP and MEP is notably stronger. This further confirms that their underperformance in our simulation study stems from the intentionally challenging evaluation conditions rather than from flawed implementation or insufficient tuning.

## C.7 ADDITIONAL RESULTS IN EXPERIMENT 2

Figure 13 shows the performance of different models across the four layouts in the user experiment. Tables 4, 5, and 6 present the statistical analyses of Experiment 2, covering the overall performance of the 4 models, their performance across different layouts, and participants' subjective ratings, respectively.

## C.8 HUMAN-SUBJECT EXPERIMENT DETAILS

We developed a web-based experimental platform with a front-end interface and deployed the RL models on a server. The front end captured participants' keypress events, which were transmitted

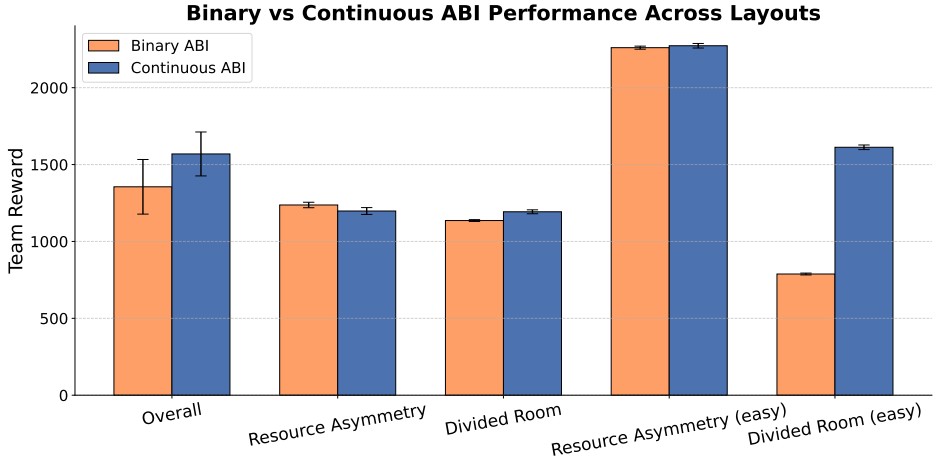

Figure 12: The comparison between continuous ABI-based TrustPOMDP and binary ABI-based TrustPOMDP. We can see that overall, continuous ABI leads to better model performance. The largest divergence appears in the *Divided Room-Easy* layout. In this layout, the continuous-ABI-based TrustPOMDP agent learns a default strategy of *waiting for help first*, whereas the binary-ABI-based TrustPOMDP agent adopts a *working alone first* strategy. The former is more appropriate, as the TrustPOMDP agent can probe the partner's trustworthiness at a relatively low time cost, enabling it to make more informed and strategically advantageous decisions in subsequent actions.

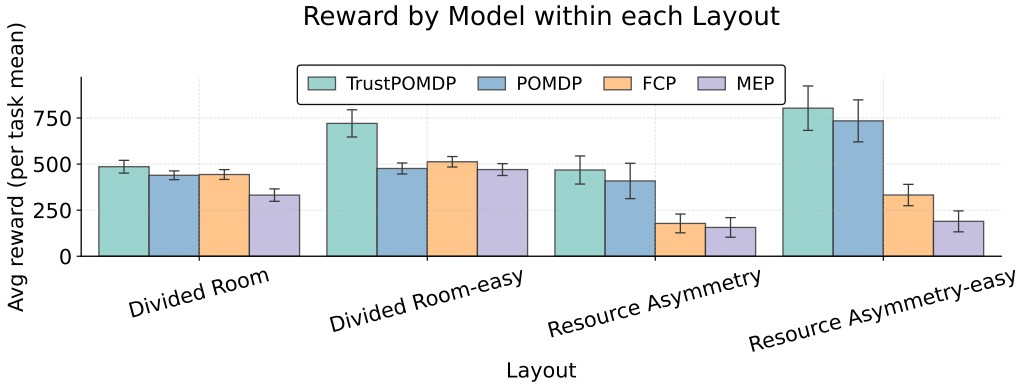

Figure 13: Detailed team performance across the four layouts in the human experiment.

Table 4: Post-hoc pairwise comparisons of overall team reward across models (Friedman test with Holm correction).

| Model A | Model B | $n$ | Mean A | Mean B | Mean Diff | Stat | $p$ | Adjusted $p$ | Effect $r$ |
|---------|---------|-----|--------|--------|-----------|------|-----|--------------|------------|
| TrustPOMDP | POMDP | 106 | 608.58 | 512.83 | 95.75 | 1606.50 | 0.0007 | 0.0013 | 0.3762 |
| TrustPOMDP | FCP | 106 | 608.58 | 375.31 | 233.27 | 1102.50 | 0.0000 | 0.0000 | 0.5305 |
| TrustPOMDP | MEP | 106 | 608.58 | 289.06 | 319.53 | 737.50 | 0.0000 | 0.0000 | 0.6423 |
| POMDP | FCP | 106 | 512.83 | 375.31 | 137.52 | 1887.50 | 0.0566 | 0.0566 | 0.2902 |
| POMDP | MEP | 106 | 512.83 | 289.06 | 223.77 | 1275.50 | 0.0000 | 0.0000 | 0.4776 |
| FCP | MEP | 106 | 375.31 | 289.06 | 86.25 | 1272.50 | 0.0001 | 0.0002 | 0.4785 |

Table 5: Pairwise reward comparisons within each layout (Friedman test with Holm correction).

| Layout | Model A | Model B | $n$ | Mean A | Mean B | Mean Diff | Stat | $p$ | adjusted-$p$ | Effect $r$ |
|--------|---------|---------|-----|--------|--------|-----------|------|-----|--------------|------------|
| Divided Room | TrustPOMDP | POMDP | 36 | 485.56 | 439.22 | 46.33 | 170.00 | 0.0483 | 0.1449 | 0.4268 |
| | TrustPOMDP | FCP | 36 | 485.56 | 443.28 | 42.28 | 166.00 | 0.1713 | 0.3426 | 0.4373 |
| | TrustPOMDP | MEP | 36 | 485.56 | 331.58 | 153.97 | 103.50 | 0.0009 | 0.0055 | 0.6009 |
| | POMDP | FCP | 36 | 439.22 | 443.28 | -4.06 | 154.50 | 0.2693 | 0.3426 | 0.4674 |
| | POMDP | MEP | 36 | 439.22 | 331.58 | 107.64 | 162.00 | 0.0342 | 0.1369 | 0.4478 |
| | FCP | MEP | 36 | 443.28 | 331.58 | 111.69 | 107.00 | 0.0019 | 0.0097 | 0.5918 |
| Resource Asymmetry | TrustPOMDP | POMDP | 22 | 467.82 | 408.41 | 59.41 | 81.50 | 0.1440 | 0.2880 | 0.3115 |
| | TrustPOMDP | FCP | 22 | 467.82 | 177.95 | 289.86 | 51.00 | 0.0127 | 0.0636 | 0.5226 |
| | TrustPOMDP | MEP | 22 | 467.82 | 156.27 | 311.55 | 36.00 | 0.0022 | 0.0132 | 0.6264 |
| | POMDP | FCP | 22 | 408.41 | 177.95 | 230.45 | 74.00 | 0.0883 | 0.2648 | 0.3634 |
| | POMDP | MEP | 22 | 408.41 | 156.27 | 252.14 | 64.00 | 0.0425 | 0.1700 | 0.4326 |
| | FCP | MEP | 22 | 177.95 | 156.27 | 21.68 | 99.00 | 0.5663 | 0.5663 | 0.1903 |
| Divided Room-easy | TrustPOMDP | POMDP | 22 | 720.64 | 476.00 | 244.64 | 27.00 | 0.0021 | 0.0126 | 0.6887 |
| | TrustPOMDP | FCP | 22 | 720.64 | 512.50 | 208.14 | 44.00 | 0.0074 | 0.0303 | 0.5710 |
| | TrustPOMDP | MEP | 22 | 720.64 | 470.18 | 250.45 | 42.00 | 0.0061 | 0.0303 | 0.5849 |
| | POMDP | FCP | 22 | 476.00 | 512.50 | -36.50 | 51.00 | 0.0142 | 0.0427 | 0.5226 |
| | POMDP | MEP | 22 | 476.00 | 470.18 | 5.82 | 106.00 | 0.7412 | 0.7412 | 0.1419 |
| | FCP | MEP | 22 | 512.50 | 470.18 | 42.32 | 38.00 | 0.0682 | 0.1364 | 0.6126 |
| Resource Asymmetry-easy | TrustPOMDP | POMDP | 26 | 803.23 | 734.27 | 68.96 | 157.50 | 0.6475 | 0.6475 | 0.0897 |
| | TrustPOMDP | FCP | 26 | 803.23 | 332.12 | 471.12 | 51.00 | 0.0009 | 0.0037 | 0.6201 |
| | TrustPOMDP | MEP | 26 | 803.23 | 189.27 | 613.96 | 26.00 | 0.0000 | 0.0002 | 0.7447 |
| | POMDP | FCP | 26 | 734.27 | 332.12 | 402.15 | 56.50 | 0.0025 | 0.0075 | 0.5927 |
| | POMDP | MEP | 26 | 734.27 | 189.27 | 545.00 | 30.00 | 0.0001 | 0.0003 | 0.7247 |
| | FCP | MEP | 26 | 332.12 | 189.27 | 142.85 | 97.00 | 0.0463 | 0.0927 | 0.3910 |

via HTTP to the server; the server processed the inputs, updated the environment state, and returned the rendered state to the front end.

At the beginning, we introduced the purpose of the study and asked participants to sign a consent form. They were then directed to an introduction page, where the task was explained. Participants were required to practice until they successfully completed one dish delivery, ensuring that they had mastered the basic gameplay before proceeding. On the instruction page, we emphasized that participants did not need to pursue the optimal strategy and could play however they preferred. This design choice was made to avoid participants' behaviors becoming overly narrow or optimized for high scores, which would reduce the effectiveness of testing model cooperation with diverse human strategies. Importantly, participants were not asked to adopt any predefined personas; they were free to play according to their own preferences.

Table 6: Pairwise comparisons of subjective questionnaire ratings across models (Friedman test with Holm correction).

| Question | Model A | Model B | $n$ | Mean A | Mean B | Mean Diff | Stat | $p$ | adjusted-$p$ | Effect $r$ |
|---|---|---|---|---|---|---|---|---|---|---|
| I Understand Agent | TrustPOMDP | POMDP | 106 | 5.21 | 5.11 | 0.09 | 1068.50 | 0.5072 | 0.5072 | 0.5409 |
| | TrustPOMDP | FCP | 106 | 5.21 | 4.23 | 0.98 | 630.50 | 0.0000 | 0.0000 | 0.6750 |
| | TrustPOMDP | MEP | 106 | 5.21 | 3.79 | 1.42 | 235.00 | 0.0000 | 0.0000 | 0.7961 |
| | POMDP | FCP | 106 | 5.11 | 4.23 | 0.89 | 423.50 | 0.0000 | 0.0000 | 0.7384 |
| | POMDP | MEP | 106 | 5.11 | 3.79 | 1.32 | 337.00 | 0.0000 | 0.0000 | 0.7649 |
| | FCP | MEP | 106 | 4.23 | 3.79 | 0.43 | 696.00 | 0.0198 | 0.0397 | 0.6550 |
| Agent Understands Me | TrustPOMDP | POMDP | 106 | 5.01 | 4.50 | 0.51 | 680.50 | 0.0144 | 0.0289 | 0.6597 |
| | TrustPOMDP | FCP | 106 | 5.01 | 3.60 | 1.41 | 531.00 | 0.0000 | 0.0000 | 0.7055 |
| | TrustPOMDP | MEP | 106 | 5.01 | 3.30 | 1.71 | 175.00 | 0.0000 | 0.0000 | 0.8145 |
| | POMDP | FCP | 106 | 4.50 | 3.60 | 0.90 | 702.00 | 0.0001 | 0.0004 | 0.6531 |
| | POMDP | MEP | 106 | 4.50 | 3.30 | 1.20 | 502.00 | 0.0000 | 0.0000 | 0.7144 |
| | FCP | MEP | 106 | 3.60 | 3.30 | 0.30 | 861.50 | 0.0756 | 0.0756 | 0.6043 |
| Agent's Adaptability | TrustPOMDP | POMDP | 106 | 5.19 | 4.31 | 0.88 | 675.50 | 0.0001 | 0.0002 | 0.6613 |
| | TrustPOMDP | FCP | 106 | 5.19 | 3.42 | 1.76 | 371.50 | 0.0000 | 0.0000 | 0.7543 |
| | TrustPOMDP | MEP | 106 | 5.19 | 3.30 | 1.89 | 246.00 | 0.0000 | 0.0000 | 0.7927 |
| | POMDP | FCP | 106 | 4.31 | 3.42 | 0.89 | 743.00 | 0.0005 | 0.0009 | 0.6406 |
| | POMDP | MEP | 106 | 4.31 | 3.30 | 1.01 | 825.50 | 0.0000 | 0.0001 | 0.6153 |
| | FCP | MEP | 106 | 3.42 | 3.30 | 0.12 | 1358.50 | 0.5798 | 0.5798 | 0.4522 |
| Cooperation Willingness | TrustPOMDP | POMDP | 106 | 5.29 | 4.92 | 0.38 | 917.50 | 0.0346 | 0.0691 | 0.5872 |
| | TrustPOMDP | FCP | 106 | 5.29 | 3.88 | 1.42 | 467.00 | 0.0000 | 0.0000 | 0.7251 |
| | TrustPOMDP | MEP | 106 | 5.29 | 3.54 | 1.75 | 188.00 | 0.0000 | 0.0000 | 0.8105 |
| | POMDP | FCP | 106 | 4.92 | 3.88 | 1.04 | 635.00 | 0.0000 | 0.0000 | 0.6737 |
| | POMDP | MEP | 106 | 4.92 | 3.54 | 1.38 | 476.00 | 0.0000 | 0.0000 | 0.7223 |
| | FCP | MEP | 106 | 3.88 | 3.54 | 0.34 | 1203.50 | 0.0870 | 0.0870 | 0.4996 |
| Satisfaction | TrustPOMDP | POMDP | 106 | 5.05 | 4.57 | 0.48 | 785.00 | 0.0160 | 0.0319 | 0.6277 |
| | TrustPOMDP | FCP | 106 | 5.05 | 3.58 | 1.47 | 421.50 | 0.0000 | 0.0000 | 0.7390 |
| | TrustPOMDP | MEP | 106 | 5.05 | 3.30 | 1.75 | 185.00 | 0.0000 | 0.0000 | 0.8114 |
| | POMDP | FCP | 106 | 4.57 | 3.58 | 0.99 | 611.00 | 0.0001 | 0.0002 | 0.6810 |
| | POMDP | MEP | 106 | 4.57 | 3.30 | 1.26 | 517.50 | 0.0000 | 0.0000 | 0.7096 |
| | FCP | MEP | 106 | 3.58 | 3.30 | 0.27 | 823.50 | 0.0989 | 0.0989 | 0.6160 |
| Agent Trusts Me | TrustPOMDP | POMDP | 106 | 5.07 | 4.58 | 0.49 | 905.50 | 0.0049 | 0.0098 | 0.5908 |
| | TrustPOMDP | FCP | 106 | 5.07 | 3.85 | 1.22 | 381.50 | 0.0000 | 0.0000 | 0.7513 |
| | TrustPOMDP | MEP | 106 | 5.07 | 3.65 | 1.42 | 237.50 | 0.0000 | 0.0000 | 0.7953 |
| | POMDP | FCP | 106 | 4.58 | 3.85 | 0.73 | 641.50 | 0.0017 | 0.0050 | 0.6717 |
| | POMDP | MEP | 106 | 4.58 | 3.65 | 0.92 | 489.00 | 0.0000 | 0.0001 | 0.7184 |
| | FCP | MEP | 106 | 3.85 | 3.65 | 0.20 | 729.00 | 0.1629 | 0.1629 | 0.6449 |
| I Trust Agent | TrustPOMDP | POMDP | 106 | 4.93 | 4.54 | 0.40 | 762.50 | 0.0157 | 0.0315 | 0.6346 |
| | TrustPOMDP | FCP | 106 | 4.93 | 3.61 | 1.32 | 440.00 | 0.0000 | 0.0000 | 0.7334 |
| | TrustPOMDP | MEP | 106 | 4.93 | 3.55 | 1.39 | 385.50 | 0.0000 | 0.0000 | 0.7500 |
| | POMDP | FCP | 106 | 4.54 | 3.61 | 0.92 | 565.50 | 0.0001 | 0.0002 | 0.6949 |
| | POMDP | MEP | 106 | 4.54 | 3.55 | 0.99 | 536.00 | 0.0000 | 0.0000 | 0.7040 |
| | FCP | MEP | 106 | 3.61 | 3.55 | 0.07 | 972.00 | 0.8004 | 0.8004 | 0.5705 |

For each task, participants first entered a practice page where they could view the layout and AI teammate and engage in trial play. In the formal task phase, they were asked to describe a self-chosen persona they intended to adopt for that round, and then play 200 steps according to that persona. After completing four rounds of a task, participants were directed to a questionnaire page, where we collected their evaluations of the cooperation experience and perceptions of the AI teammate. After finishing the first task, participants proceeded to complete the remaining two tasks, following the same procedure across all 4 tasks.

# D EXPERIMENTS IN A NEW TASK ENVIRONMENT

To further evaluate the generalizability of our approach beyond the commonly used Overcooked environment, we implemented and tested our method in a coin-collection task (McKee et al., 2024).

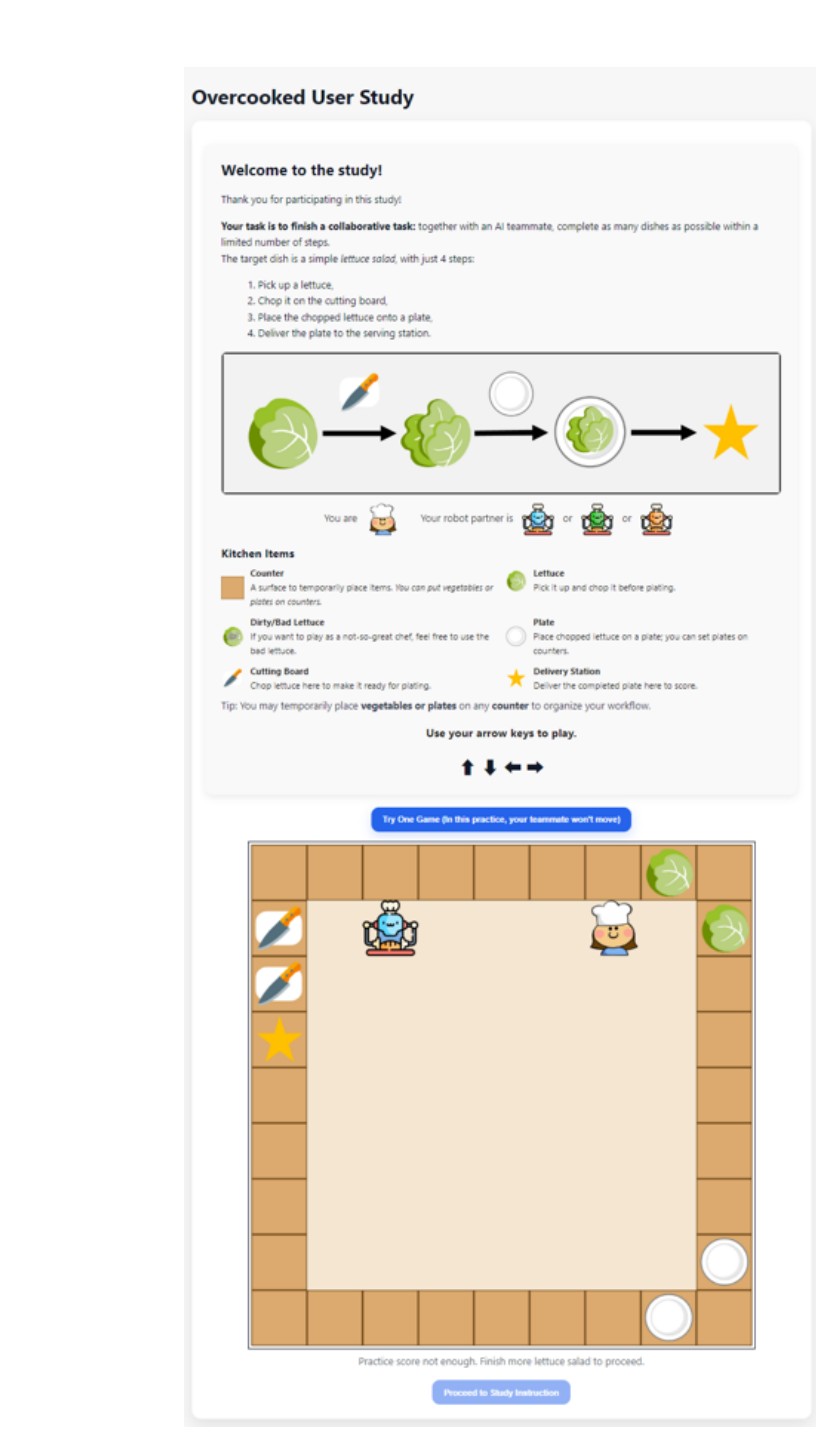

Figure 14: The introduction page in the human experiment.

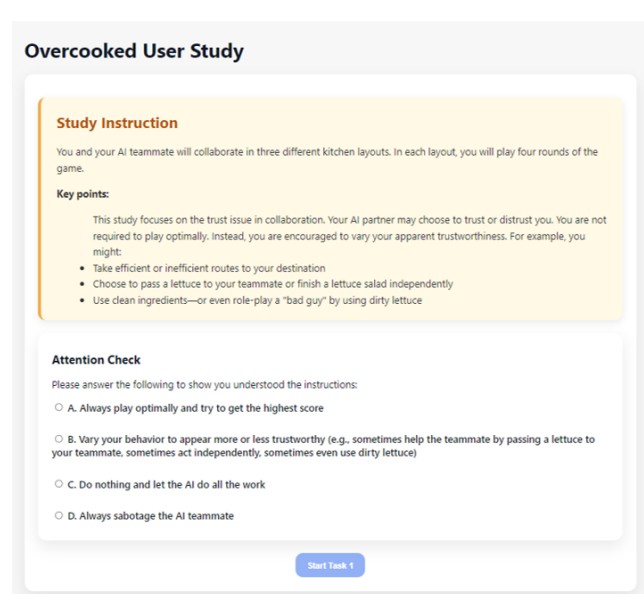

Figure 15: The task instruction page.

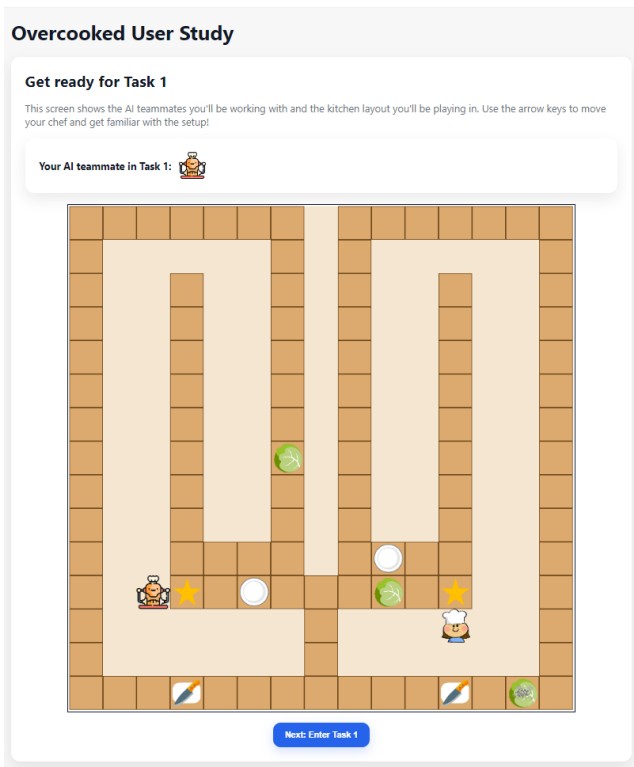

Figure 16: The practice page for a new task, where participants were introduced with the assigned agent and layout.

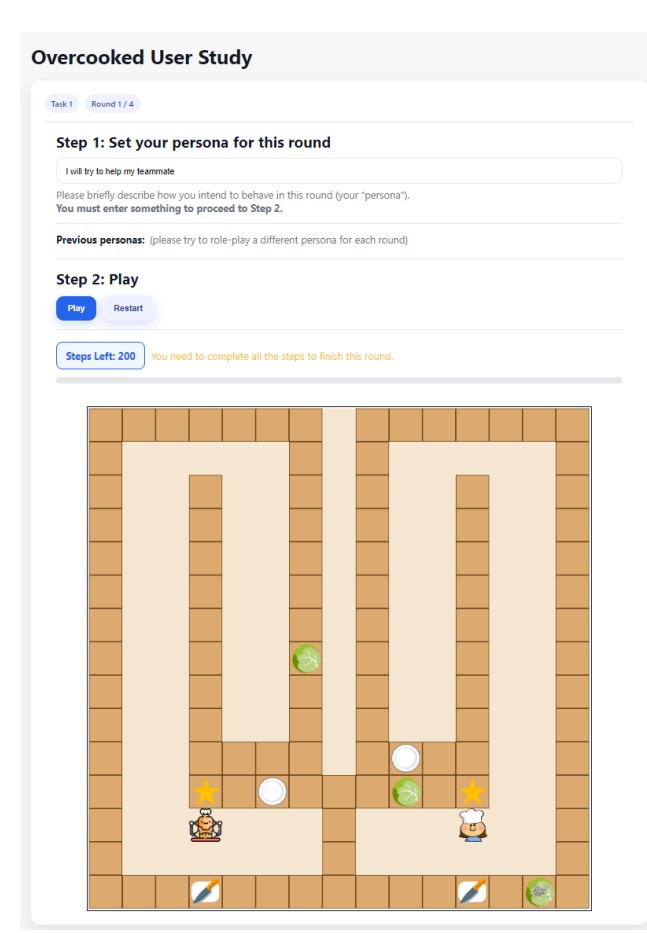

Figure 17: The main task page in the human experiment, where participants needed to first specify a persona whatever they liked to play, then played with the agent for 200 steps.

Figure 18: The questionnaire page after each task, where we collected participants' subjective ratings of different statements.

### D.1 TASK AND ENVIRONMENT DESCRIPTION

As shown in Figure 19, the environment consists of two agents (blue and red), two coins (red and blue), and wall obstacles. Agent aims to collect the coins in the environment.

#### D.1.1 ACTIONS, STATES, AND REWARD STRUCTURE

Each agent has five possible actions: move up, down, left, right, or stay still. Attempts to move into a wall result in no movement.

The state representation includes the positions of both agents, the coin positions, and wall locations. Once a coin is collected, its position is set to $(-1, -1)$.

The reward structure is described in Table 7. Collecting one's own colored coin yields a reward of $+5$ and does not affect the teammate's reward. Collecting the teammate's coin yields $+10$ to the collector but incurs a $-5$ penalty to the teammate. If the episode reaches the maximum number of steps before both coins are collected, each agent receives a penalty of $-1$, and there is an additional step penalty of $-0.1$ to encourage efficient behaviors.

This reward design induces a social dilemma: individually rational behavior (stealing the partner's coin) leads to a lower collective outcome (each agent gains only 2 points if both steal), while mutual cooperation (each collecting their own coin) yields a jointly optimal outcome (5 points each), resembling a Prisoner's Dilemma structure (Axelrod, 1980). We designate the blue agent as the Ego agent and the red agent as the partner.

#### D.1.2 PARTNER STRATEGY VARIATION VIA ABI

The partner's behavior is driven by variations in Ability, Benevolence, and Integrity:

- **Ability**: A high-ability partner moves efficiently toward goals, whereas a low-ability partner performs noisy and less rational actions.

- **Benevolence**: A high-benevolence partner considers both agents' rewards and thus collects its own coin. A low-benevolence partner prioritizes only self-reward and collects the opponent's coin.

- **Integrity**: A high-integrity partner truthfully reveals its intention through movement, whereas a low-integrity partner may initially behave cooperatively but later switch to stealing the opponent's coin, intentionally misleading the Ego agent.

Thus, the Ego agent must infer the partner's underlying ABI state and act accordingly. For example, when interacting with a high-benevolence partner, the Ego agent should collect its own coin. Against a low-benevolence partner, it should instead target the partner's coin to avoid exploitation. If integrity is low, the Ego agent must resist deception and avoid miscalibrated reliance. When a partner is high in benevolence and integrity but low in ability, the Ego agent should adapt to assist or compensate.

### D.2 ABI IMPLEMENTATION

We use the same ABI definitions as in the Overcooked experiments.

**Ability:** High ability uses a near-deterministic policy, while low ability sets the Boltzmann rationality parameter $\beta = 0.1$ in Eq. 1 to introduce randomness.

**Benevolence:** High benevolence is implemented by setting $(\alpha = 0.5, \beta = 0.5)$ in Eq. 2, weighting both agents equally. Low benevolence sets $(\alpha = 1, \beta = 0)$, considering only self-reward.

**Integrity:** We implement the $\mathcal{V}$-events in Eq. 3 using deceptive movement patterns. For example, a high-benevolence agent may initially move toward the opponent's coin (appearing selfish) before switching back to collect its own coin, thereby producing a "false-negative" intent cue. Conversely, a low-benevolence agent may initially behave cooperatively before defecting. These symmetric deception patterns constitute integrity-violation events.

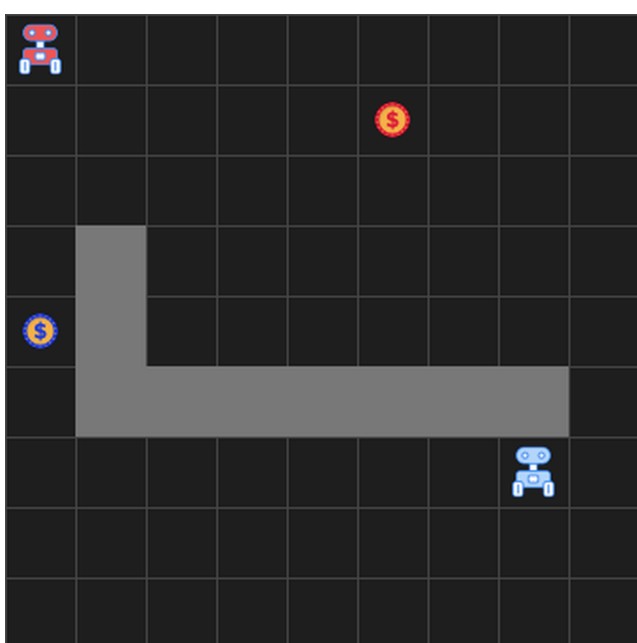

Figure 19: The coin-collection environment. There are two agents (red and blue) and two coins (red and blue) in the environment.

Table 7: Reward structure based on coin color

| Coin color | Reward for self | Reward for co-player |
|---|---|---|
| matching | +5 | +0 |
| mismatching | +10 | -8 |

### D.3    TRUSTPOMDP TRAINING

We instantiated eight types of ABI-producing partner agents corresponding to all combinations of high/low values in Ability, Benevolence, and Integrity (i.e., $2^3 = 8$). This partner population is used as the training set.

Following the same TrustPOMDP training pipeline used for Overcooked, the Ego agent trains across episodes with different partners sampled randomly from the population. Every five episodes, the partner is switched to ensure exposure to diverse behaviors. ABI inference is updated using the most recent 10-step observation history and appended to the Ego agent's observation vector.

The Ego agent receives a *team reward* and we trained the TrustPOMDP agent using Proximal Policy Optimization (PPO) with a multi-layer perceptron policy. The hyperparameters followed common best practices in deep reinforcement learning. In particular, we set the learning rate to $3 \times 10^{-4}$, the rollout length to $2048$ environment steps, and the minibatch size to $64$. Each policy update consisted of $10$ epochs of stochastic gradient descent. We used a discount factor of $\gamma = 0.99$, a GAE parameter of $\lambda = 0.98$, and an entropy coefficient of $0.2$ to encourage exploration. The clipping threshold for the surrogate objective was $0.3$, and the value-function loss coefficient was set to $0.7$. To stabilize optimization, gradient norms were clipped at $1.0$. The policy network architecture was implemented via `policy_kwargs`, as detailed in Section B.3. We trained the TrustPOMDP model for 2.5M steps.

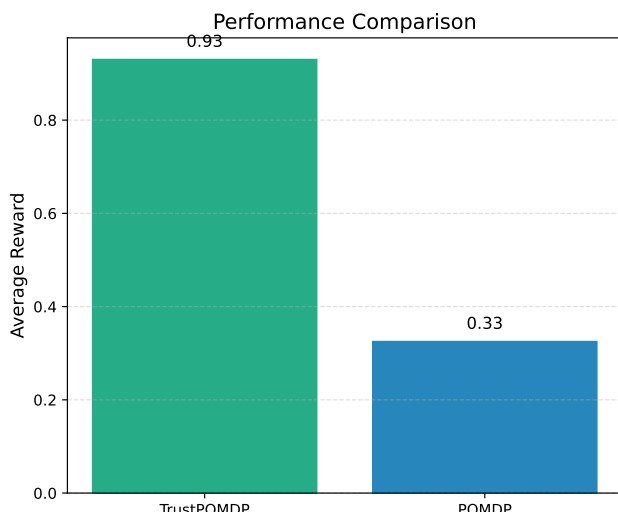

Figure 20: Performance comparison between TrustPOMDP and POMDP in the coin-collection environment. The results are tested against 8 types of partners varying along A, B, I.

### D.4  EXPERIMENTS WITH SIMULATED PARTNERS

We compared TrustPOMDP against a basic POMDP baseline trained with the same partner population (same parameters and training steps) but without ABI inference or conditioning. Results (Figure 20) show that TrustPOMDP consistently achieved higher team rewards.

We observed that both models learned a default policy of first collecting their own coin, as this yields a higher expected reward initially. However, when interacting with low-benevolence partners who steal the Ego agent's coin, TrustPOMDP successfully inferred the selfish intent after the initial interaction and switched to stealing the partner's coin. In contrast, the basic POMDP continued moving toward its own coin and typically failed before adjusting, since its partner would steal the coin first. This illustrates the benefit of our proposed ABI-guided adaptation.

We did not include FCP or MEP as baselines because their partner populations do not fully cover variability across the full ABI space, and are theoretically worse than our basic POMDP baseline.

Overall, this experiment demonstrates the generalizability of our ABI-based partner modeling and TrustPOMDP framework. The results confirm that explicitly inferring and conditioning on ABI enables more robust cooperation and dynamic adaptation in mixed-motive (even social dilemma) settings.

## E  USAGE OF LARGE LANGUAGE MODELS

We used GPT-5 to check grammar and mathematical formulas. In addition, the cartoon elements in Figure 1 were created with the assistance of GPT-5.

