# OpenReview forum: "Learning to Cooperate with Humans through Theory-Informed Trust Beliefs"
_ICLR.cc/2026/Conference — Submitted to ICLR 2026_

### Official Review · Reviewer_DLMJ · 2025-10-28

**Soundness:** 2
**Presentation:** 3
**Contribution:** 3
**Rating:** 4
**Confidence:** 4

**Summary:**

This paper tackles the coordination problem in multi-agent reinforcement learning, by borrowing from literatures on trust in human-AI teaming, and constructing a novel trust-based POMDP formalism as well as a belief inference method. Experiments are conducted both in the popular simulation task of Overcooked, and with real humans.

**Strengths:**

The modeling of ABI presents a well-thought out application of the vast literature on trust to the RL setting. The methods for ABI inference and conditional-policy optimization are kept simple yet effective. The experiments, particularly the task and partner designs, are done well, and the statistical significance is convincing.

**Weaknesses:**

My main concern for this paper is that 1) the POMDP baseline and TrustPOMDP method perform very similarly, and 2) FCP and MEP severely underperform basic POMDP.
1) Given all the added mechanisms for ABI inference and conditioning, I would expect the performance gap to be way larger.
2) It's hard to say how I would expect POMDP to compare to FCP and MEP, but I find it strange that just how severely FCP and MEP methods underperform basic POMDP. This leads me to think FCP and MEP have much left on the table in terms of hyperparameter tuning.

I would be willing to raise my score if this main concern is addressed.

**Questions:**

- line 254: should the ground truth ABI labels be in {0,1} instead of [0,1], since they have been discretized to be binary?
- Fig 5 and 7b: I recommend shifting  the X labels to the right a bit, so that they are centered with the bar plots.




On citation:
It has been a while since I studied this field, but I believe there are a few key missing works that should be cited here. This includes the seminal work in [1], a recent work that studies trust in MARL for Hanabi agents [2], and the works of [3] that studied legibility and predictability. Although [3] is in the robotics domain, legibility and predictability are still notions adjacent to trust that I believe should be included in this work.


[1] John D Lee and Katrina A See. Trust in automation: Designing for appropriate reliance. Human factors, 46(1):50–80, 2004.

[2] H. C. Siu et al., “Evaluation of Human-AI Teams for Learned and Rule-Based Agents in Hanabi,” in Advances in Neural Information Processing Systems, Curran Associates, Inc., 2021, pp. 16183–16195.

[3] A. D. Dragan, K. C. T. Lee, and S. S. Srinivasa, “Legibility and predictability of robot motion,” in 2013 8th ACM/IEEE International Conference on Human-Robot Interaction (HRI), Mar. 2013, pp. 301–308. doi: 10.1109/HRI.2013.6483603.

---

> ### Author Response · Authors · 2025-11-27
> **Author Rebuttal**
>
> (1/2)
>
> We thank the Reviewer for the thoughtful feedback and for recognizing that our modeling of ABI represents a well-considered application of the extensive trust literature to the RL setting. We also appreciate the Reviewer’s important concerns regarding model performance, which help to further clarify and strengthen our evaluation.
>
> ---
> **Performance gap between TrustPOMDP and POMDP**:
> - We thank the Reviewer for raising this important concern. We agree that, given the additional ABI inference and conditioning mechanisms, one might expect a substantially larger performance gap between TrustPOMDP and the POMDP baseline. The observed similarity can be explained by three factors.
> - First, TrustPOMDP’s performance gains stem from two complementary sources: (i) exposure during training to partners with diverse ABI profiles and (ii) the explicit ABI inference and conditioning mechanism. While the latter is essential, the former also contributes substantially and constitutes a key methodological contribution of our work. As a result, even the plain POMDP baseline benefits from training with diverse partner behaviors, which naturally narrows the performance gap.
> - Second, the main paper reports performance averaged across four layouts. As shown in Figure 9 (the original manuscript) and discussed in Sec. 5.1 and Appendix A.3, TrustPOMDP’s advantages are most pronounced in the “ambiguity zone,” where the partner’s intentions cannot be reliably inferred from immediate observations and must instead be reasoned through ABI belief updates. We deliberately included both high-ambiguity and low-ambiguity layouts to transparently demonstrate when ABI inference provides substantial benefits. Consequently, TrustPOMDP significantly outperforms POMDP in high-ambiguity layouts, while differences are marginal in low-ambiguity ones.
> - Third, we found during revision that although the layouts contained ambiguity zones, their proportion relative to the full 400-step episode was small, limiting the overall impact of trust calibration on aggregate performance. To address this, **we have conducted new experiments with shorter episodes, increasing the relative presence of ambiguity zones**. The new results show a more pronounced performance gap, with TrustPOMDP demonstrating clearer advantages over POMDP (see Figures 5 and 8 in the revised manuscript).
> - Overall, these findings align with our theoretical expectation that the benefits of ABI inference emerge most clearly in contexts with sustained ambiguity that require belief-based reasoning. We have clarified this relationship in the revised manuscript and included new experimental results to better substantiate this effect.

---

> ### Author Response · Authors · 2025-11-27
>
> (2/2)
>
> ---
> **Performance of FCP and MEP**:
> - We thank the Reviewer for raising this important concern. We first would like to clarify that our implementations of FCP and MEP are faithful to their original papers. We closely followed the reported implementation details and hyperparameter settings, and additional tuning attempts did not yield substantial performance changes. Moreover, by examining the self-play (SP) agents used to construct partner populations used in training FCP and MEP, we confirmed that they exhibit reasonable task performance, indicating that the observed results are not due to implementation flaws or degenerate training.
> - The primary cause of the performance gap lies in the nature of the partner populations used by FCP and MEP. Both methods rely on SP-trained agents as partners, which, due to decentralized training, tend to solve tasks independently and avoid cooperative behaviors, resulting in consistently low Benevolence. In contrast, real humans often show an intrinsic willingness to help, and our ABI-producing agents explicitly cover systematic variations in Benevolence. Similarly, SP agents are trained with team rewards and rarely exhibit low-Integrity behaviors (e.g., using dirty lettuce), whereas real humans may occasionally violate optimal or normative behavior. Our ABI-producing agents explicitly model such variability, capturing a broader and more realistic range of partner behaviors. As a result, SP-based populations cluster around low Benevolence and high Integrity.  Although FCP and MEP introduce diversity along Ability via using different training checkpoints, they fail to capture complex mixed-motive scenarios involving Benevolence and Integrity.
> - A secondary factor is the nature of our simulation study, which intentionally includes rule-based and out-of-distribution (OOD) partner behaviors as stress tests for robustness. Under these extreme conditions, FCP and MEP perform particularly poorly in certain layouts. For instance, in the Resource Asymmetry layout, where agents share access to global items and interference is stronger, FCP- and MEP-based agents struggle more when facing OOD behaviors. In contrast, in the Divided Room layout, where agents operate more independently, their performance remains relatively stable. Importantly, in our human study with more realistic participant behaviors, the performance of FCP and MEP is notably stronger, confirming that their underperformance in the simulation experiment reflects the deliberately challenging evaluation conditions rather than flawed implementation or insufficient tuning.
> - Overall, the observed performance gap primarily reflects the limitations of SP-based partner populations in capturing the diversity of human cooperative behavior, rather than weaknesses in implementation or hyperparameter selection. **We have added an analysis of the performance of FCP and MEP in the revised manuscript** (see Appendix C.6, Page 23).
> ---
> **Typos**:
> - We thank the Reviewer for pointing this typo out. **We have fixed this in the revised manuscript**.
> ---
> **Figure x-axis labels**:
> - Thank for the comments! **We have adjusted the figures**.
> ---
> **Citation Missing**:
> - We thank the Reviewer for pointing out these important and highly relevant works. **We have added all the suggested references to the revised manuscript**.

---

### Official Review · Reviewer_Qs1v · 2025-11-01

**Soundness:** 2
**Presentation:** 2
**Contribution:** 3
**Rating:** 4
**Confidence:** 3

**Summary:**

The paper looks at the problem of incorporating trust considerations into the decision-making process of an AI agent. In particular, they take into account the ABI model and create a latent variable that corresponds to each dimension of this model in a POMDP representation of the agent decision-making problem. The values for these dimensions are inferred on the fly as the agent interacts with the human. The validity of their method is tested using both simulated experiments and with real user study and the initial results seems promising.

**Strengths:**

I believe the inclusion of concepts and insights from social science is a worthwhile pursuit. In regard to such efforts, starting from well-defined frameworks like ABI makes sense to me. I also appreciate the fact that the authors actually ran user studies to validate their proposed model.

**Weaknesses:**

Unfortunately, I have quite a few concerns about the current approach.

POMDP - First of all, POMDPs are not a good framework to capture human-AI interaction or multi-agent interactions in general. As the authors point out, while there are works that try to leverage POMDPs, these methods are far too limited and cannot capture many of the more important aspects of the human-AI interaction problem. For example, how humans would be actively reasoning about what the agent might be doing and trying to adapt to it. Also, the fact that the agent and human could have independent objectives, and in some cases, objectives that might conflict with each other. As such, I would recommend that the authors look at more general frameworks like I-POMDPs [1].

Assumptions about human inference - The method currently assumes that the method by which humans infer the ABI parameter is known upfront, and additionally that you can convert these dimensions into simple parameters of a model. While I am aware of some psychological evidence for the noisy rational model (though even that is known to fail at times), I didn't see any discussions about the other choices. Also, for the beta-distribution, the current reference points to a paper that shows that the distribution could conceptually be used to capture trust accumulation. However, I didn't see any user study data in that paper (maybe I missed it), which suggests that it is a descriptive model insofar as it can simulate how people's trust evolves.

Benevolence - Isn't the formulation of benevolence in individual time-steps a bit simplistic? As an extreme example, consider an agent that is following a policy that is optimal for both reward functions, versus one where the agent is going out of its way to help the human. In terms of perception, wouldn't the latter be perceived as being benevolent? However, in your current reward wheight scheme they might not be differentiated.

Minor Issue: Equation 7 seems to be incomplete

[1] Gmytrasiewicz, Piotr J., and Prashant Doshi. "Interactive POMDPs: Properties and preliminary results." International Conference on Autonomous Agents: Proceedings of the Third International Joint Conference on Autonomous Agents and Multiagent Systems-. Vol. 3. 2004.

**Questions:**

I would appreciate if the authors could respond to each of the points raised in the weakness section

---

> ### Author Response · Authors · 2025-11-27
> **Author Rebuttal**
>
> (1/2)
>
> We thank the Reviewer’s insightful feedback and for recognizing that incorporating concepts from social science is a worthwhile direction. We also appreciate the Reviewer’s identification of the missing theoretical grounding and discussion. In response, we have strengthened the paper’s theoretical foundation by clarifying its connection to the I-POMDP framework and expanding the discussion of the ABI inference mechanism.
>
> ---
> **I-POMDPs framework**:
> - We sincerely thank the Reviewer for this constructive feedback.
> - We carefully read the I-POMDPs paper as suggested by the Reviewer. The core idea of I-POMDPs is that an agent's belief is defined as a probability distribution over both the environmental states and the models of other agents. In this paper, our approach can be viewed as a simplified instantiation of the I-POMDP framework, in which the AI agent maintains beliefs over the state of the environment and the model of its human partner, represented by ABI (Ability, Benevolence, and Integrity), and makes decisions based on these beliefs.
> - Moreover, I-POMDPs offer a flexible framework for recursive belief modeling: not only can agent *i* update its belief about agent *j*, but agent *j* can, in principle, also update its belief about agent *i*. In such fully recursive settings, agent *i* would need to anticipate how agent *j* updates its beliefs in response to observed behaviors.
> - In this work, however, we focus on a single-sided belief formulation, where the AI agent models the human partner but does not explicitly model the human's belief about the AI. This design choice is driven by our goal of creating a human-belief-agnostic collaborative agent, one that can robustly adapt to diverse human partners without relying on assumptions about their internal beliefs regarding the AI. Such a formulation better reflects real-world settings, where human beliefs are highly heterogeneous and often unobservable. We consider bidirectional belief modeling an important direction for future work.
> - **We have added I-POMDPs to the revised Section 3 (Preliminaries) and explicitly explained the relationship between our approach and the I-POMDP framework, highlighting its theoretical grounding**.
> - Finally, we view the explicit modeling of humans’ beliefs about agents (i.e., bidirectional belief nesting) as an exciting and important direction for future research, and we plan to explore this extension in our future work.
> ---
> **Discussion of other choices beyond noisy rational model**:
> - We thank the Reviewer for this thoughtful comment. We believe there may be a misunderstanding regarding the role of the ABI model in our framework, and we appreciate the opportunity to clarify this point.
> - The ABI models do not describe how humans infer ABI. Instead, they are behavior-generating models to generate agent policies that mimic how humans may act along the Ability, Benevolence, Integrity. With the ABI models, we can train a population of partner agents with different ABI. The trained partner agents are further used to train TrustPOMDP and POMDP.
> - Meanwhile, we agree with the Reviewer’s comments that other design choices for generating ABI-related behaviors are possible. In the revised manuscript, **we have added an explicit discussion of alternative approaches for generating ABI-characterized behaviors and their associated trade-offs, clarifying the rationale behind our design decisions (see Section 4.2 Line 227-235)**.
> ---
> **Beta-distribution for representing trust**:
> - We appreciate the Reviewer for pointing this out. We use Beta-distribution due to two main advantages. First, it limits an estimation interval to 0 and 1, creating consistency with the trust measurement scale. Second, this model accounts for a historical representation by accumulating the number of successful and unsuccessful collaborations. These advantages make beta representation suitable for the probabilistic estimation of trust.
> - We apologize that the cited literature doesn’t contain empirical human data. **We have added more empirical literature to support this choice in the revised manuscript** (see Sec. 4.3).
> - Also, to show the effectiveness of the Beta-distribution based trust representation, in the revised manuscript **we have added an explicit empirical analysis of ABI inference dynamics**, illustrating how our Beta-distribution based model captures trust evolution (see Figures 6 and 11). Specifically, we visualize how the inferred ABI values evolve over time in response to partner behaviors, demonstrating that the current ABI inference model can produce meaningful and interpretable trust accumulation trajectories.

---

> > ### Author Response · Authors · 2025-11-27
> >
> > (2/2)
> >
> > **Simplistic formulation of Benevolence**:
> > - We agree with the Reviewer that the current formulation of Benevolence is a bit simplistic.
> > - In our current formulation, Benevolence is defined through the weighting between self-reward and other-reward. We choose this straightforward formulation because in our setting, the two agents’ rewards don’t overlap, so the current simple formulation doesn’t undermine our current results.
> > - However, I understand the Reviewer’s point that if the two agents’ rewards overlap, it is hard to use current formulation to distinguish between (1) working for shared rewards and (2) working only for others’ reward.
> > - Thanks to the Reviewer’s suggestion, in our revised manuscript, we improved the Benevolence formulation to a more nuanced mechanism (see Sec 4.2 Line 235-247), which includes three components: self-reward, other-reward, shared-reward. The formulation could be:
> > - R = α * self-reward + β * other-reward + (1 - α - β) * shared-reward.
> > - where self-reward denotes the reward obtained exclusively by the agent itself, other-reward denotes the reward obtained exclusively by the partner agent, and shared-reward denotes the reward jointly shared by both agents. The parameters α and β lie in [0,1] and satisfy α + β ≤ 1.
> > - In this work, we treat both increasing the partner’s reward and increasing the shared reward as manifestations of benevolence. Therefore, benevolence can be effectively controlled using a single parameter α. A larger α indicates a more self-oriented, low-benevolence agent, whereas a smaller α implies that the agent places greater emphasis on the partner’s or shared reward, corresponding to a high-benevolence agent.
> > - This improved formulation enables a more nuanced representation that better accounts for reward overlap and aligns more closely with perceptual interpretations of benevolence.
> > ---
> > **Typos**:
> > - We thank the Reviewer and have fixed the typo.

---

### Official Review · Reviewer_vNUL · 2025-11-01

**Soundness:** 3
**Presentation:** 2
**Contribution:** 2
**Rating:** 4
**Confidence:** 4

**Summary:**

The paper deals with a very important research line and makes a valuable contribution towards cooperative AI through grounding in social theory and providing evidence of improved human-AI coordination. The integration of psychological constructs into a POMDP is novel and practically relevant. And their "TrustPOMDP" in general is shown to yield favourable results.

**Strengths:**

-This is a very important research line; we need more and more research and investigation in Human-AI cooperation.

-The method draws on well-established social science theory, so-called the Ability, Benevolence, and Integrity (ABI) model of trust (Mayer et al., 1995). This connection bridges human behavioral theory and reinforcement learning (RL), offering conceptual interpretability uncommon in multi-agent learning.

-The paper also introduces  TRUSTPOMD, an extension to the standard POMDP by incorporating a latent trust-belief model, with only minimal additional variables per dimension,  yet still meaningfully alters the agent behavior.

**Weaknesses:**

-We need to see stronger evidence of generality beyond the commonly used "overcooked environment" as this is the only environment that the paper uses.  Also a clearer empirical comparison to baselines is required.  The authors did not include the ablated POMDP model in the evaluation with human participants but I would have liked to see, e.g.,  the effect of ABI inference on the subjective perceptions of human participants. As it stands now, it is difficult to argue that a plain POMDP model would not have achieved the same results in the human evaluation.

Other suggestions to improve paper:

-Consider including one additional cooperative task (e.g., social dilemmas) to demonstrate domain generalization.

-Consider adding more complex examples of norm violations to test integrity, such as a partner who occasionally cheats or lies about completing a task. This would show how well the model handles deceptive behavior.

**Questions:**

Q1:Could you share any results or arguments that show that the ABI inference itself (rather than general POMDP adaptation) is what drives the higher trust and satisfaction reported by participants?

Q2:How does the model handle cases where a human partner changes behavior midway through a task, for example, becoming less cooperative or breaking norms?

Q3:Do you have any results as to what happens when you represent ABI as continuous values instead of binary ones?

---

> ### Author Response · Authors · 2025-11-27
> **Author Rebuttal**
>
> We thank the Reviewer’s thoughtful feedback and for recognizing the value of our work in investigating human–AI cooperation. We also appreciate the important comments regarding the missing experimental analysis. In response, we have rerun all experiments to directly address these concerns and strengthen the empirical evidence.
>
> ---
>
> **Comparison between TrustPOMDP and POMDP in the human evaluation**:
>
> - We thank the Reviewer for this important point. In the original submission, to prevent participant fatigue, we compared TrustPOMDP only against established baselines and did not include its ablated POMDP version in the human evaluation.
> - To directly address this concern, **we have conducted a new user study that includes the ablated POMDP model alongside TrustPOMDP**. The new results show that TrustPOMDP leads to significantly higher team reward and better user experience compared to the plain POMDP, providing empirical evidence that the ABI inference mechanism itself plays a critical role in shaping participants’ subjective perceptions.
> - **We have incorporated these new experimental results into the revised manuscript (see the revised Section 6.2 and Figure 8)**.
>
> ---
>
> **Midway behavior change handling**:
> - We appreciate the Reviewer for this insightful suggestion.
> - In the revised paper, **we have explicitly evaluated this scenario by introducing mid-task behavior transitions in two directions: from cooperative to uncooperative (good to bad) and from uncooperative to cooperative (bad to good)**. We conduct these tests across all three ABI dimensions. **We have provided some examples in the revised manuscript** (see Figures 6 and 11).
> - In the provided cases, we manipulate the partner agent’s benevolence, including four situations: “consistently low’’, “consistently high’’, “from low to high’’, “from high to low’’. We visualize the temporal dynamics of ABI inference and corresponding behavioral adaptation of the TrustPOMDP agent. The results show that our ABI inference model can effectively detect these mid-task changes, and the TrustPOMDP agent can adjust its behavior in response to the updated ABI belief.
> - Overall, these findings demonstrate that both the ABI inference module and the TrustPOMDP framework are capable of robustly handling dynamic partner behavior and adapting to partners’ behavior shifts during ongoing interactions.
> ---
> **Comparison between continuous ABI and binary ABI**:
> - We thank the Reviewer for this valuable question. In the previous version, for simplicity, we just used binary representation of ABI. To address this concern, **we have trained a TrustPOMDP model with continuous ABI and compare it to TrustPOMDP with binary ABI**.
> - Our comparative results (see Appendix C.5 and Figure 12 in the revised manuscript) show that, overall, continuous ABI leads to slightly better performance compared to binary ABI. Especially in the Divided Room-Easy layout, continuous ABI-based TrustPOMDP is significantly better than binary ABI-based TrustPOMDP (and we also analyzed the reason). It is not surprising since the continuous variables have greater representational capacity, which allows the model to capture more fine-grained variations in partner behavior and trustworthiness.
> - Since the continuous ABI yields overall better performance and to address other concerns raised by the reviewers, **we have rerun both the simulation and user studies using a continuous ABI-based TrustPOMDP. The updated results indicate that TrustPOMDP remains superior to the baseline methods including its ablated version** (see Figures 5 and 8 in the revised manuscript).
> ---
> **Test generalizability in new environment**:
> - To directly address Reviewer's concern, we have incorporated a social dilemma environment [1] in which two colored agents collect two colored coins within a limited time. When an agent collects a coin which matches its own color, it receives a positive reward without affecting its partner. However, collecting a color-mismatched coin grants the collector a higher reward while reducing the partner’s reward.
>
> - This setup mirrors a Prisoner’s-Dilemma–style social dilemma: cooperation is mutually beneficial if both agents cooperate, but defection becomes individually rational if the partner defects.
>
> - Following the Reviewer’s suggestion, we operationalize Integrity as cheating behavior: an agent with low integrity initially behaves cooperatively but later switches to collecting the partner-colored coins, thereby misleading the teammate and violating expectations.
>
> - **We have implemented and evaluated TrustPOMDP in this new environment. The results show that TrustPOMDP again outperforms the basic POMDP and better adapts to different partner types. Full details of the environment and experiments are provided in Appendix D of the revised manuscript**.
>
> [1] McKee, K. R., Bai, X., & Fiske, S. T. (2024). Warmth and competence in human-agent cooperation. Autonomous Agents and Multi-Agent Systems, 38(1), 23.

---

### Author Response · Authors · 2025-12-04
**To AC: Summary of Responses**

Dear AC,

We sincerely appreciate your additional efforts during this challenging review cycle and your valuable contributions to the ICLR community. To facilitate your assessment, we have prepared a clear summary of the reviewers’ concerns and our corresponding revisions.

---

## Overview of the Paper

This work addresses a central challenge in human-AI collaboration:

> How can an AI agent effectively collaborate with diverse human partners whose behavior and reliability vary?

Grounded in established social science trust theory, we propose that the AI agent should be able to decide whether and when to trust the human. We operationalize the Ability–Benevolence–Integrity (ABI) trust model, enabling the agent to infer human latent trustworthiness and adapt its reliance accordingly.

We introduce TrustPOMDP, where belief over human ABI is explicitly incorporated into the agent’s observation space and used for policy adaptation. We evaluate in simulation and a 106-participant human study, and find that TrustPOMDP significantly outperforms strong baselines.

All reviewers appreciated the strong theoretical grounding and highlighted multiple positive aspects.
- vNUL described it as "a very important research line" with conceptual interpretability uncommon in MARL
- DLMJ & Qs1v praised the simplicity and effectiveness of TrustPOMDP
- Qs1v emphasized the solid integration of simulation and human experiments
- DLMJ commended the quality of our experimental design

Reviewers also raised serveral concerns and suggestions — all now fully addressed.

---

## Major Reviewer Concerns and Our Revisions

**(1) Performance comparison with POMDP (Reviewers vNUL and DLMJ)**

Reviewer vNUL pointed out the lack of a direct TrustPOMDP vs. POMDP comparison in the human experiment. Originally, we compared TrustPOMDP to POMDP and other established baselines in simulation, but only to established baselines in the human study. Reviewer DLMJ also noted that the performance gap between TrustPOMDP and POMDP appeared not that big. We clarified that the key advantage of TrustPOMDP emerges in “ambiguity zones,” where the partner’s intention cannot be inferred from immediate observations. Our original episodes were long, so ambiguity zones occupied only a small portion of each episode, making the advantage appear less pronounced.
To address both concerns, we shortened episode length, retrained all models, and reran both the simulation and human experiments with POMDP included in the human study. The new results show that TrustPOMDP consistently and significantly outperforms POMDP across both experiments. Updates are provided in Sec. 5 and Sec. 6.

**(2) Mid-task behavior changes and trust accumulation dynamics (Reviewers vNUL and Qs1v)**

Reviewer vNUL requested evaluation of mid-task behavior changes, and Reviewer Qs1v asked for clarification on how the Beta-based trust model supports trust accumulation. In the revision, we explicitly test transitions in both directions: cooperative → uncooperative and uncooperative → cooperative, across all three ABI dimensions. We visualize temporal ABI inference and the TrustPOMDP agent’s behavior under four representative conditions: consistently low, consistently high, low→high, and high→low. Results (Figures 6 and 11) show that TrustPOMDP effectively detects changes and adapts its behavior in a timely and appropriate manner, fully addressing both concerns.

**(3) Theoretical grounding and modeling discussion (Reviewer Qs1v)**

We added a detailed connection to the Interactive-POMDP literature (Sec. 3), clarifying conceptual grounding. We expanded the discussion of our Beta-distribution–based ABI inference mechanism (Sec. 4.3), and analyzed its empirical effectiveness (see response above). We further added discussion of alternative modeling choices for each ABI dimension and clarified why we chose the current design (Sec. 4.2). We also refined the definition of Benevolence (revised Eq. 2). Additional references were incorporated.

**(4) Generalizability in a new social-dilemma environment (Reviewer vNUL)**

To further demonstrate generalizability, we implemented an additional experiment in a new coin-collection social dilemma environment. We also modeled Integrity as deceptive behavior. Results (Appendix D) show that TrustPOMDP continues to adapt effectively to different partner types, reinforcing the robustness of our method.

**(5) Performance of FCP and MEP (Reviewer DLMJ)**

We added a detailed analysis explaining why FCP and MEP underperform compared to POMDP and TrustPOMDP (Appendix C.6).

**(6) Minor issues**

We added missing related work, improved figure quality, and corrected remaining typos.

---

We sincerely thank the AC and all reviewers. The revised manuscript substantially strengthens both theoretical framing and experimental rigor, and directly addresses every concern raised. We hope you will consider this strengthened version for acceptance at ICLR.

---

### Meta-Review · Area_Chair_89GX · 2026-01-08

**Summary:**

The paper addresses the challenge of integrating trust factors into the decision-making processes of an AI agent. Specifically, the authors utilize the ABI model, developing a latent variable for each dimension of this model within a partially observable Markov decision process (POMDP) framework to represent the agent's decision-making dilemma. As the agent engages with humans, the values for these dimensions are dynamically inferred. The effectiveness of their approach is evaluated through both simulated experiments and real user studies, yielding initial results that appear promising.

**Reviewer Concerns:**

The experiment settings are limited, for example, more environment should be evaluated, more clearer empirical comparison is needed.

**Reviewer Scores:**

The scores of the reviewers are 4,4,4, which are consistent and suggest that the paper is below the bar of ICLR.
POMDP, which is selected by the authors, could be limited in capturing human-AI interaction or multi-agent interactions.

---

### Decision · Program_Chairs · 2026-01-26

Reject